
**Creating a national scale debris flow susceptibility model for Great Britain:**
**a GIS–based heuristic approach**
Emma J. Bee[1], Claire Dashwood[1], Catherine Pennington[1], Roxana L. Ciurean[1] and Katy Lee[1]
[1] British Geological Survey, Environmental Science Centre, Nicker Hill, Keyworth, UK.
*Correspondence to*: BGS Enquiries (enquires@bgs.ac.uk)
**Abstract.** Debris flows in Great Britain have caused damage to transport infrastructure, buildings, and disruption
to businesses and communities. This study describes a GIS–based heuristic model developed by the British
Geological Survey (BGS) to produce a national scale spatial assessment of debris flow susceptibility for Great
Britain. The model provides information on the potential for debris flow occurrence using properties and
characteristics of geological materials (permeability, material availability and characteristics when weathered),
slope angle and proximity to stream channels as indicators of susceptibility. Building on existing knowledge, the
model takes into account the presence or absence of glacial scouring. As determined by the team of geologists
and geomorphologists, the model ranks the availability of debris material and slope as the two dominant factors
important for potential debris flow initiation, however it also considers other factors such as geological controls
on infiltration. The resultant model shows that over 90 % of the mapped debris flows in the BGS inventory
occurred in areas with the highest potential for instability and approximately 6 % were attributed to areas where
the model suggested that debris flows are unlikely or not thought to occur. Model validation in the Cairngorm
Mountains indicated a better performance, with 93.50 % in the former and less than 3 % in the latter category.
Although the quality of the input datasets and selected methodological approach bear limitations and introduce a
number of uncertainties, overall, the proposed susceptibility model performs better than previous attempts,
representing a useful tool in the hands of policy-makers, developers and engineers to support regional or national
scale development action plans and disaster risk reduction strategies.
**1 Introduction**
The term debris flow refers to the rapid downslope flow of poorly-sorted debris mixed with water (Ballantyne
2004). Debris flows are described by (Hungr et al. 2014) as "*very rapid to extremely rapid surging flows of*
*saturated debris in a steep channel*". They are a widespread phenomenon in mountainous terrain and are distinct
from other types of landslides, as they can occur periodically on established paths, usually gullies and first- or
second-order drainage channels and are characterised by "*strong entrainment of material and water from the flow*
*path*" (Hungr et al. 2014). Debris flows consist of three main parts: source area, track and depositional area. Source
areas may be initiated by a slide, debris avalanche or rock fall from a steep bank, or by spontaneous instability of
the steep stream bed (Hungr et al. 2014). Irrespective of the mode of flow initiation, debris flows are generated
when a build-up of pore water pressures in unconsolidated sediments causes a reduction in the shear resistance,
leading to failure and sediment flow (Ballantyne, 2004). Debris flows tend to follow long, narrow tracks. The
upper, erosional section of the flow consists of a gully that is continued downslope by parallel levées of dominantly
coarse debris that enclose the track of the flow (Fig. 1) and often terminate downslope in one or more lobes of
bouldery debris (Ballantyne 2004). Characteristic morphological features used to distinguish debris flow fans
from other sediment-laden process depositional areas include: high slope angle of the fan, very large individual





particles, coarse levées and boulder trains, signs of impact loading on obstacles, U-shaped eroded channels and
steep, debris-loaded channels upstream (Hungr et al. 2014).

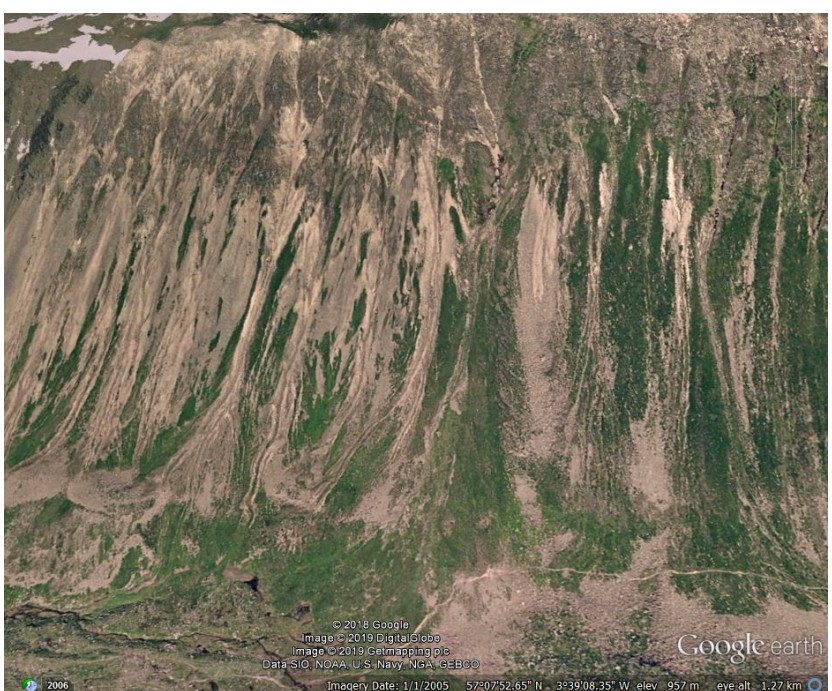

**Figure 1: Debris flows in Lairig Grhu, Cairngorms, Scotland with distinct levée features (Image Source: Google Earth.**
**Digital Globe 2019)**
Debris flows in Great Britain are most commonly found in upland Scotland but also in parts of Wales and the
Lake District, England. According to Nettleton et al. (2005), Ballantyne (2004), and Cruden (1996) there are two
types of debris flow in Great Britain: hillslope or open slope debris flows and valley-confined or channelised
debris flows. *Hillslope* or *open slope debris flows* (Fig. 2a) form their own path down the valley slopes as tracks
or sheets and deposit material on the lower slopes where the gradient shallows. *Valley-confined* or *channelised*
*debris flows* (Fig. 2b) originate in bedrock gullies and are confined for at least part of their length along the gully
floor. The flows have the consistence equivalent to that of wet concrete and can be fronted by a boulder
concentration or 'head'. The two categories are transitional; many valley-confined flows debouch on to open
ground in their lower reaches, and hillslope debris flows often follow shallow gullies cut in valley-side drift, talus
or regolith. Most debris flows in Great Britain occur following a period of high magnitude precipitation events.





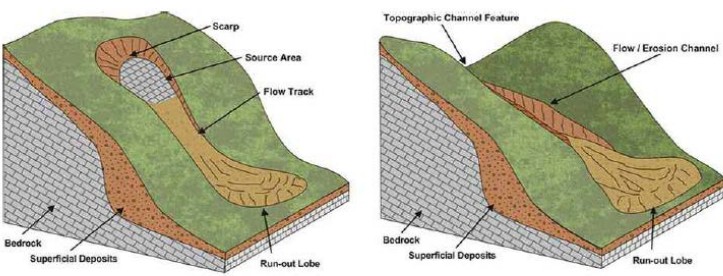

**Figure 2: Hillslope (a) and channelised (b) debris flow (Image source: Nettleton et al. 2005)**

### 1.1 Rationale for research

Debris flows are potentially very destructive as they can cause significant erosion of the substrates over which they flow, thereby increasing their sediment charge and further increasing their erosive capabilities (Nettleton et al. 2005). Debris flows can lead to financial loss for anyone involved in the ownership or management of property, including developers, householders, loss adjusters, surveyors or local government. These costs could include increased insurance premiums, depressed house prices and, in some cases, engineering works to stabilise land or property.

In Great Britain, the Scottish road and rail networks are recurrently affected by debris flows. In August 2004, two debris flows intersected the A85 in Glen Ogle, north of Lochearnhead, Stirlingshire. Fifty seven people were stranded on the roadway between two debris flows with a cumulative volume of approximately 15,000 $m^3$ (Winter et al. 2014) and either left the scene on foot or were rescued by helicopter (Milne et al. 2009). The A85, which normally carries up to 5,600 vehicles per day, was closed for four days (Winter et al. 2006). The most widely reported location in Great Britain for debris flow impact on a strategic road is the A83 'Rest and Be Thankful' Pass (British Geological Survey 2009). Event magnitudes here are generally small, ranging between 200 and 1,000 $m^3$ in volume, however debris flows have occurred at least on an annual basis over the last 25 - 30 years (Winter et al. 2014). The road is regularly closed in both directions resulting in a 55 mile diversion with significant regional economic impact that is regularly reported in the media. Postance et al. (2017) calculated that historic estimates of the economic impact of the 2007 A83 '*Rest and Be Thankful*' debris flow event totalled £1.2 million over a 15 day closure, 60 % greater than previous estimates. In 2011 and 2014, wig-wag warning signs (Winter et al. 2013, Winter and Shearer 2017) and ten bespoke debris flow barriers, respectively, were installed to warn drivers about the increased likelihood of debris flows (Maccaferri 2014).

The potential risk to people, business and properties outlined in the England and Wales Planning Policy Guidance Note 14 (PPG14) and associated annexes (Department of the Environment 1990) were the main drivers for the development of British Geological Survey's (BGS) slope instability datasets, including the current Debris Flow Susceptibility Model (DFSM). Although this guidance was intended for England and Wales only, the principles are relevant to Scotland as well (Jones and Lee 1994). PPG14 has now been replaced by the National Planning Policy Framework, 2012 (Department for Communities and Local Government 2012), however, unstable land still requires consideration and PPG14 remains the only document widely available that gives any

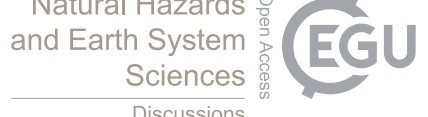

guidance for planning. To overcome this limitation and respond to the needs of asset managers, decision-makers
and practitioners, a methodology was developed to assess debris flow susceptibility spatially at national scale.
The aim was to identify potential debris flow initiation areas and serve as an indication where further, more
detailed studies should be carried out.

5        Debris flow hazard assessment methodologies vary widely depending on the purpose of the analysis, the extent

of the study area and data availability. They are often divided into two phases 1) the identification of potential
sources or initiation areas and, 2) the estimation of the runout, the former being the focus of the present study.
While statistical models for potential source identification are based on extensive inventories of past events
(Blahut et al. 2010, Carrara et al. 2008), deterministic approaches consider physical characteristics of the process
and are thus transferable to any site (Iovine et al. 2003) but are high data demanding and require calibration, which
makes them rarely feasible for national scale applications. Geographical Information System (GIS) based
statistical landslide susceptibility assessments are focused, more often than not, on the tool rather than the input
data, and do not distinguish between landslide type, resulting in an oversimplification of the landslide controlling
factors (Van Westen et al., 2006). Heuristic methods, on the other hand, link a variety of environmental factors
contributing to possible slope instabilities to expert knowledge of the area and can be adapted to any scale. In this
study, we opted for the latter approach, as it allows for the combination of uniform geologic datasets available at
a broad scale with an indepth local expert knowledge.

18       As to the scale of analysis, most debris flow susceptibility assessments are carried out at local, catchment or

regional scales (e.g. Hurlimann et al. 2008, Kappes et al. 2011, Skinner 2013, Blais-Stevens and Behnia 2016).
Few countries or jurisdictions have developed guidelines on how to map and assess debris flow hazard, with the
exception of Austria and Switzerland, who have legislated debris flow management since 1975 (Jakob 2005), and
Scotland, since 2005 (Winter et al. 2005).

23       In Great Britain, studies have concentrated on debris flow susceptibility modelling and hazard ranking in

Scotland only. The first regional debris flow susceptibility assessment was the Scottish Road Network Landslides
Study (Winter et al. 2005). This was commissioned by the Scottish Executive and conducted by a multidisciplinary
working group in response to debris flow events in 2004 that impacted Scotland's road network substantially. A
pan-Scotland susceptibility assessment was carried out within a GIS environment (Harrison et al. 2006), which
considered availability of material, water conditions, vegetation and land cover, proximity of stream channels and
slope angle as preconditioning factors. The assessment was calibrated by the working group and then interpreted
to derive hazard and hazard-ranking information for the Scottish road network (Winter et al. 2013). Areas of
England and Wales also prone to debris flows, such as the Lake District and Snowdonia National Park, were
excluded from this assessment. When the methodology used in the Scottish Road Network Landslide Study was
applied to the rest of Great Britain, it was apparent that the model was over sensitive in many areas where debris
flows are not common. These erroneous results led to its reassessment for application at the national scale. The
subsequent availability of the national Soil Parent Material Map produced by BGS (Lawley et al. 2009) enabled
the use of more detailed information to determine the character and availability of regolith than in the previous
2005 assessment.



**2 Data and methods**
In order to develop a national scale debris flow susceptibility model for Great Britain, the characterisation of
the geological and geomorphological factors that increase the likelihood of debris flow occurrence and their
potential to be represented spatially within a GIS was explored. To assess the performance of the model, aerial
photographs and LiDAR imagery were used to create an inventory of over 2000 debris flows. The spatial location
of the perceived debris flow initiation area was represented by a point recorded in GIS and subsequently compared
with the susceptibility model pixel classification at that location. The model accuracy was assessed using a
frequency ratio plot and the Receiver Operating Characteristics (ROC) curve in a representative area for debris
flow occurrence. ROC curves are commonly used to evaluate the performance of binary classifiers i.e. presence
or absence of landslides. Since landslide inventories are rarely complete, the tool was tested in an area where most
debris flows had been mapped. The methodological workflow is illustrated in Fig. 3 and explained in the following
sections.

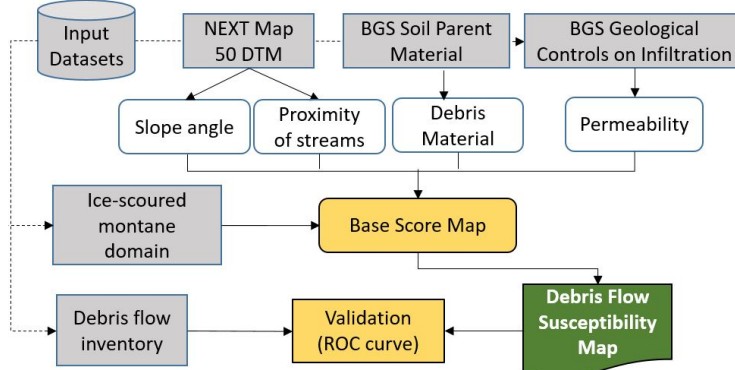

**Figure 3: Methodological workflow.**
**2.1 Factors that increase the likelihood of debris flows**
An analysis of a number of debris flow susceptibility maps by Carrara *et al.* (2008) showed that, despite a
large number of variables being used, only a few had a strong discriminant power: high slope angle, pasture or no
vegetation cover, availability of detrital material and active erosional processes. This echoes previous research,
undertaken as part of a study into debris flows affecting the Scottish transport network (Winter et al. 2005), that
identified five main factors to be considered when determining the hazard potential for debris flows:
1. availability of debris material
2. hydrogeological conditions
3. slope angle
4. proximity of stream channels
5. land use.
All above factors were considered in creating the BGS DFSM for Great Britain (version 6.0), with the
exception of land use. Whilst vegetation may have a beneficial effect on slope stability (e.g. intercepting rainfall,
removing soil moisture and reinforcement of the ground through root networks) the amount of stabilisation will


vary with the type of vegetation and the season. Experience working on a number of projects with the Forestry
Commission highlighted the fact that, even though an area may be designated as woodland, it is not always
completely planted and forest roads and firebreaks may increase the potential for debris flows. For this reason,
the fact that land use changes over time, and the scale to which the model was being developed, the authors
excluded land use from the model, focussing on the geological and morphological factors that contribute to debris
flow initiation. It is recommended that local knowledge and up to date, detailed land use information is used by
end users to support the modelling results.
**2.2 Predisposing morphological and geological factors**
For each of the factors that increase the likelihood of a debris flow occurring, a spatial dataset was created to
indicate where these factors were most and least prominent. These factors are described in the following sections.
**2.2.1 Availability of Debris Material**
Research on debris flows in Scotland has shown that failures are most likely on slopes mantled by regolith or
coarse-grained (cohesionless) superficial deposits with a sandy matrix (Ballantyne 2004). Granular materials are
more susceptible to debris flows due to higher infiltration rates and greater potential for rapid increase in pore
pressures during intense rainfall events (McMillan et al. 2005). The methodology adopted for assessing
availability of material sought to classify geological materials according to texture and the characteristics of any
weathering products (regolith) that may be mantling slopes and that could become involved in a debris flow.
Formations prone to this include granite and sandstones as opposed to finer-grained schistose or extrusive igneous
lithologies (Milne 2008).
In order to create a spatial data layer that classifies geology based on its susceptibility to debris flow
occurrence, the BGS Soil–Parent Material Database (version 6) (Lawley et al. 2009) was analysed. A 'soil parent
material' is a geological deposit over, and within which, a soil develops (Lawley et al. 2009). Typically, the
parent material is the first recognisable geological deposit encountered when excavating beneath the soil layer. It
represents the very-near-surface geology. In general, the geological deposits closer to the ground surface are the
most weathered, whilst the deeper deposits are less so. Soil parent materials play a vital role in determining soil
type as their characteristics control three primary properties of their overlying soils: chemistry, texture and
permeability–porosity (drainage). The latter two are key controls of the propensity of a material to fail as a debris
flow. A GIS based logical decision-tree algorithm was developed, using expert knowledge, to determine the
propensity, on a scale from 1 (low susceptibility) to 10 (high susceptibility), of each soil–parent material to fail as
a debris flow. Thickness of material (and therefore source availability) was inferred by expert geologists through
this scoring process. The algorithm considered the substrate of the geological material (i.e. whether it was bedrock,
superficial or a surficial deposit), its origin (i.e. igneous, sedimentary or metamorphic) and the parent-materials
strength characteristics when weathered to produce a map showing the propensity for the parent material at any
given location, to fail as a debris flow (Fig. 4).




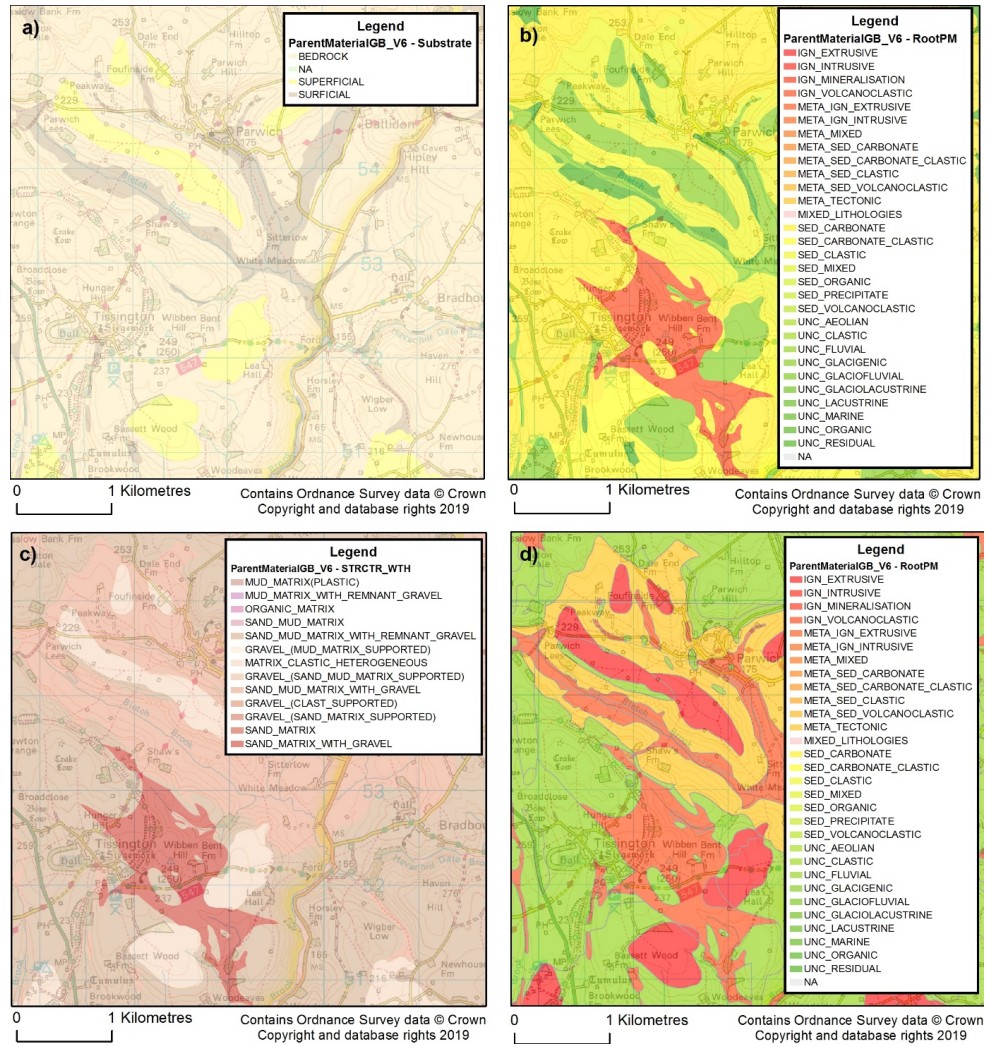

**Figure 4: Excerpt of the Soil–Parent Material Spatial Database, when coded to reflect a) substrate b) origin of parent material and c) characteristics of material when weathered. d) The propensity of the material to fail as a debris flow based on expert judgement.**

### 2.2.2 Hydrological conditions (Permeability)

Debris flows are usually triggered by intense precipitation events. Harrison et al. (2006) used two criteria when determining whether a material was more or less susceptible to debris flow occurrence due to its hydrogeological characteristics: a) the ability of water, as rainfall or overland flow, to infiltrate a potentially mobile deposit *(permeability of the deposit)* and b) the ability of water to remain within the deposit to an extent where pore water pressures can build to a level where the shear strength is sufficiently reduced to initiate failure *(permeability of the underlying material)*.





If a potentially mobile deposit is permeable but the underlying deposit is of a more impermeable nature,
infiltration of water will be impeded and this can lead to an increase in pore water pressure, subsequent lowering
of shear strength and potential failure. Conversely, if the underling material is permeable, water flow will not be
impeded and a rapid increase in pore water pressures during an intense rainfall event is less likely.
In order to assess the geological factors controlling the hydrological conditions of the ground, BGS has
developed a national scale 'Geological Controls on infiltration dataset' (GCI) (Mee et al. 2016) for internal use.
This dataset gives an indication of how easily water can penetrate into the ground and describes whether: a)
infiltration is likely to be controlled by superficial or bedrock permeability, or both; and b) the infiltration
conditions are likely to be free draining, highly variable or poorly draining.
Infiltration rates are dependent on the thickness of any superficial deposits present, which in turn determines
whether infiltration is primarily controlled by the permeability of the bedrock or the superficial layers, or a
combination of the two. The GCI dataset is based partly on the methodology used to create the 'drainage' layer
of the Infiltration Sustainable Drainage Systems (SuDS) GIS dataset, where superficial and bedrock lithologies
are scored from 1 to 3 according to their infiltration capacity (Dearden 2016, Dearden et al. 2013). The GCI dataset
is incorporated into the model by adapting the infiltration score to reflect their potential impact on controlling
debris flow occurrence (Table 1). The infiltration scores are then modified to reflect a 10-point scale, as indicated
in Table 1 and Fig. 5.
**Table 1: Infiltration score within the Geological Controls on Infiltration GB dataset (V7) and their relevance to the**
**potential debris flow occurrence.**

| Score | Description | Relevance | Reclassified score |
|-------|-------------|-----------|--------------------|
| 1 | The subsurface is likely to be free draining | The infiltration conditions are such that, depending on the presence or not of other determining factors (e.g. slope, characteristics of geological material.) the potential for debris flows is low | 1 |
| 2 | The subsurface is likely to have highly variable drainage | The infiltration conditions are such that, depending on the presence or not of other determining factors (e.g. slope, characteristics of geological material etc.) the potential for debris flows is moderate | 5 |
| 3 | The subsurface is likely to be poorly draining | The infiltration conditions are such that, depending on the presence or not of other determining factors (e.g. slope, characteristics of geological material etc.) the potential for debris flows is high | 10 |




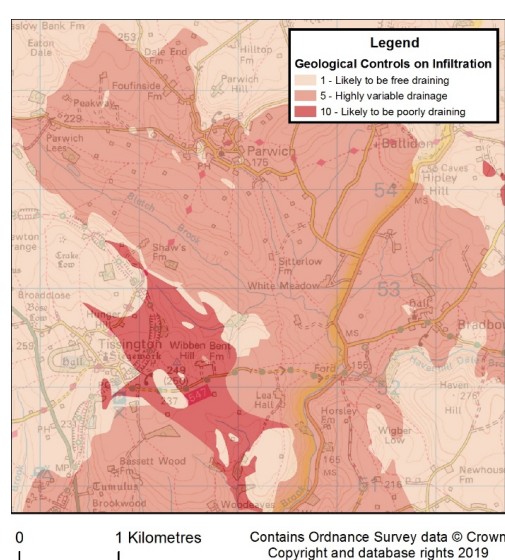

**Figure 5: Excerpt of the reclassified Geological Controls on Infiltration dataset.**

**2.2.3 Slope angle and proximity of stream channels**

A key control on debris flow initiation is slope angle. Debris flows generally have a minimum limiting angle of around 30° but can be initiated on gully floors on slopes as low as 15 - 20° (Innes 1983). Ballantyne (2004) states that "*surveyed hillslope flows in Scotland have source areas on slopes of 30 - 46°, with most starting on gradients of 32 - 42°*". This range concurs with other published studies on debris flow initiation (Ballantyne 2004, Innes 1983, Milne 2008, Winter et al. 2005). Innes (1983) and Milne (2008) observe that channelised debris flows can initiate on lower angle slopes and, as such, channelised debris flows should be modelled differently to open hillslope debris flows. Furthermore, Heald and Parsons (2005) and Innes (1983) identify that the maximum slope angle for debris flow initiation is between 46 - 50°, since above this gradient, debris can no longer accumulate. Using the information acquired in previous studies and observations, Table 2 indicates the scores assigned to each slope angle category. Scores assigned to slopes overlaying a stream channel were increased by a factor of two in the 20 - 30° and 15 - 20° categories to denote their higher potential for debris flow initiation than open hillslopes with equal gradients. To obtain the slope angle and stream channel network a 50 m Digital Terrain Model (DTM) derived from the NEXTMap™ data was used. Flow direction, flow accumulation and network were modelled using the archyrdo tool in ArcGIS ESRI. An excerpt of the slope dataset (with stream channels) and the associated scores are shown in Table 2 and Fig. 6 respectively.

**Table 2: Scores assigned to slope angle. Score increased to 16 or 28 if slope angle located within a stream channel.**

| Slope angle (°) | 0-3 | 3-15 | 15-20 | 20-30 | 30-32 | 32-42 | 43-46 | 46-50 | >50 |
|---|---|---|---|---|---|---|---|---|---|
| Score | 0 | 2 | [1]4 | [2]6 | 8 | 10 | 8 | 6 | 2 |



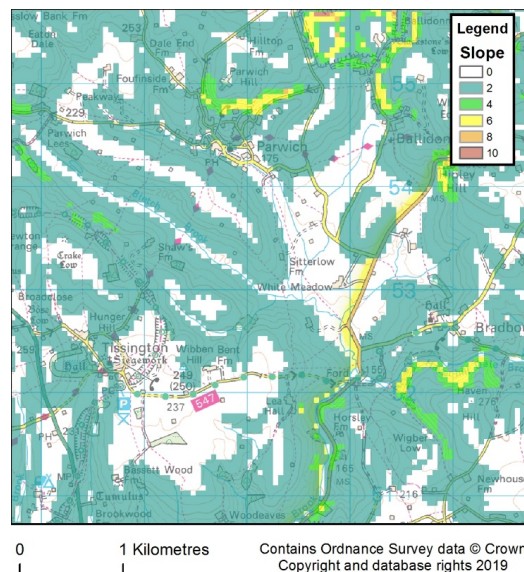

**Figure 6: Excerpt of the slope dataset Slope categorisation dataset with stream channels, derived from the NEXTMap**
**Britain elevation data from Intermap Technologies. Categorised scores as shown in Table 2.**
The NEXTMap™ Britain data that was used in the slope and channel assessment is an elevation product
generated by Intermap Technologies in 2005 using an X-band interferrometric synthetic aperture radar system
(IFSAR) mounted on an airborne platform. The original Digital Elevation Model (DEM) contains all artefact
features such as buildings and wooded areas. The algorithms that were employed to produce the Digital Terrain
Model (DTM) were generally very effective. However, some areas of woodland, particularly those on slopes, are
identified as areas of higher declivity than in reality, and thus negatively influence the modelling output at those
locations. The same applies to steep sided edges of quarries and coastal features (Fig. 7). Although these issues
result in localised modelling errors, the NEXTMap™ DTM is considered to be, in general, an accurate dataset.
Most importantly, it provided a continuous coverage of Great Britain and was deemed to have the best available
and accessible data at the time of production.




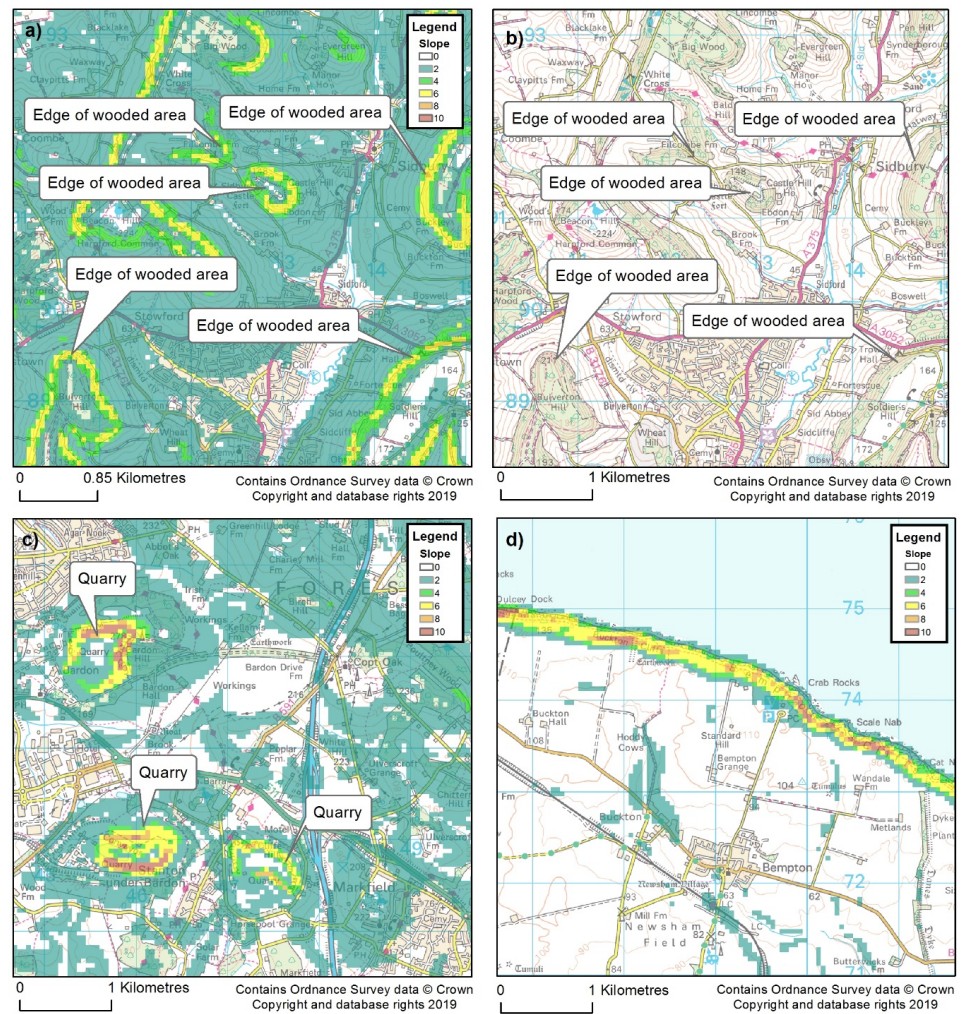

**Figure 7: Slope categorisation dataset, derived from the NEXTMap Britain elevation data from Intermap Technologies, indicating modelling artefacts. Upper figures (a and b) highlight the effect of wooded areas; lower figures (c and d) indicate slopes associated with quarrying and coastal features.**

**2.2.4 Glacial Scouring**

According to Ballantyne (2004), debris flows are scarce in areas of extensive glacial scouring such as the Outer Hebrides, Knoydart, Morven and Argyll. This observation is supported by the analysis of the aforementioned debris flow inventory created for model validation. The dataset didn't contain any recorded debris flows in areas of extensive glacial scouring (Fig. 8b).

In order to reflect the impact of ice scouring on reducing the likelihood of debris flows in affected areas of North West Scotland, the BGS Quaternary Domain Map (Booth et al., 2012) is used (Fig. 8a). Herein, the 'ice-scoured montane' domain is defined as largely devoid of superficial deposits and having experienced severe,

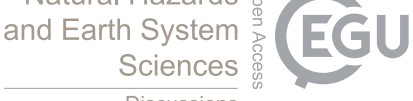



widespread glacial erosion resulting in very thin or non-existent soil with minimal occurrence of deeply weathered
bedrock. It can be expected that in these areas, there is less material available for debris flows to occur. To ensure
that the areal extent of this dataset matched the resolution and extent of the more modern BGS data being included
in the model, aerial imagery interpretation and expert judgment were employed to produce a more spatially and
geologically accurate, rather than cartographic, 'ice-scoured montane' domain output (Fig. 9).

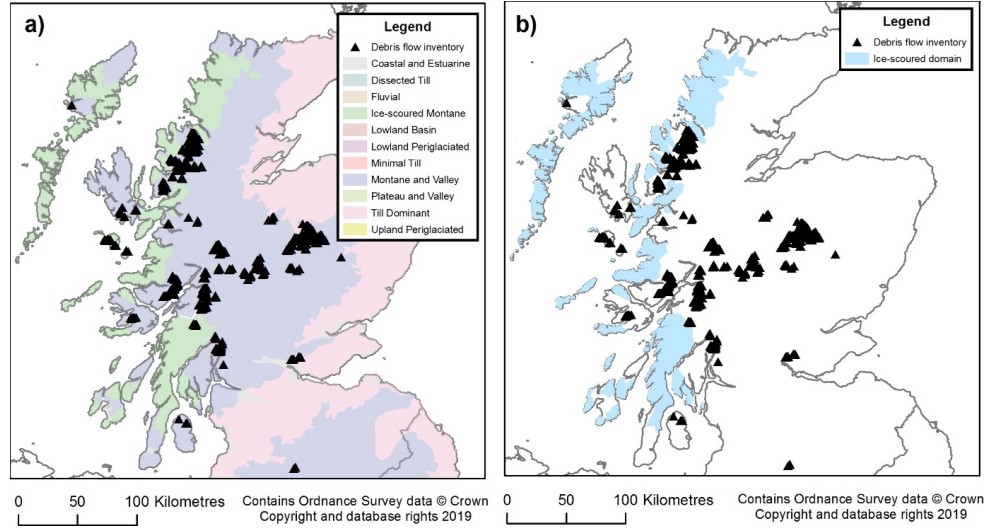

**Figure 8: a) The 'cartographic' BGS Quaternary domains map and debris flow inventory showing the 'ice-scoured**
**montane' domain. b) The revised ice-scoured montane domain map used within the debris flow model.**
**2.3 Model application**
Having identified, created and classified the datasets that reflect the geological and morphological factors
increasing the likelihood of debris flow occurrence based on a ten-point scoring scale, the next step was to combine
them based on their relative importance. Each dataset was converted into raster format using a 50 m cell size and
applying the maximum value rule, whereby the cell's value reflects the maximum value of the data it overlaid,
irrespective of size of coverage within that cell. Expert opinion amongst the team of geologists developing the
model determined 'availability of debris' and 'slope' as the two most important and thus dominant factors, with
geological controls on infiltration being relatively less important for potential debris flow initiation. To convey
this, the model generated a product of the 'availability of debris material' and 'slope' factors and then added the
permeability score (Equation 1). This means that, in order to be assigned a high susceptibility base score, a pixel
must have a high score for availability of debris material as well as slope angle. Where two pixels have the same
multiplied value, the permeability score is used to further differentiate between them (Eq. (1)). Conversely, if, for
example, permeability scored a maximum value without a significant slope or available material score, a debris
flow is unlikely to be initiated.

Susceptibility Base Score = (Debris Material Score × Slope Angle Score ) + Permeability Score    (**Eq. 1**)





Once the debris flow susceptibility base score was determined for each pixel using Equation 1, the ice scoured
montane domain mask was applied to the resulting map. The values of those pixels within the ice scoured montane
domain were divided by three to reduce their influence in the overall model; this value was selected after some
trial and error and comparison against known areas of debris flow occurrence. Table 3 indicates the categories
used to classify the final debris flow susceptibility scores into five classes, A to E, and their associated description.
In order to define the class boundaries, two experts independently assigned boundaries by assessing all possible
combinations produced by the components (parent material, slope and permeability) and their scorings (1-10).
They then came together to discuss the decisions that they had made and using scenarios (i.e. thinking about where
you might find a location where the potential for the parent material to fail was scored as 4, but slope was a 10
and permeability was a 10) came to a consensus on where the final A-E class boundaries should be placed.
**Table 3: Final debris flow susceptibility classes scoring and description.**

| Score | Legend | Interpretation | Description |
|---|---|---|---|
| 0 - 10.99 | A | Debris flows are not thought to occur. This is due to a lack of available slope materials, high drainage rates or low slope angle. | Debris flows are not thought to occur |
| 11 - 32.99 | B | Debris flows are not likely to occur. This is either due to a limited availability slope materials, sufficient drainage rates or low slope angles. | Debris flows are not likely to occur |
| 33 - 49.99 | C | Debris flows may be present or anticipated. The combinations of increasing slope angle, poor drainage condition and the presence of available material may increase the potential for failures to occur. | Debris flows may be present or anticipated |
| 50 - 64.99 | D | Debris flows are probably present or have occurred in the past. The combinations of steep slopes, poor drainage conditions and an increased presence of available material suggest that debris flows are likely to be present at these sites. | Debris flows are probably present or have occurred in the past |
| 65 - 110 | E | Debris flows are highly likely to be present. The heightened combinations of steep slopes, poor drainage conditions and the presence of available material suggest that debris flows are highly likely to be present at these sites. | Debris flows are highly likely to be present |

**3 Results and discussion**
The resultant debris flow susceptibility map for Great Britain (Fig. 9) is a 50 m raster based GIS dataset which
provides information on the potential for debris flow initiation at a given location (Bee et al. 2017a and Bee et al.
2017b). The susceptibility model is classified in a five-point scale from A (i.e. debris flows are not thought to
occur) to E (i.e. debris flows are highly likely to be present) and covers Great Britain, but excludes the Isle of
Man, the Channel Islands and Northern Ireland. Table 4 shows how the model compares against all debris flow
points registered in the inventory. Over 90 % of the recorded debris flows occurred within categories D and E,
that is, areas with the highest potential for instability. However, approximately 6 % of the mapped debris flows
were attributed to category A or B, which, according to the model, would have suggested that debris flows are
unlikely or not thought to occur.




1  **Table 4: Comparison of debris flow model for Great Britain (v6.0) against mapped debris flow occurrences (n = 2087).**

| Susceptibility class | Number of observed debris flows | % of observed debris flows |
|---|---|---|
| A | 1 | 0.05 |
| B | 124 | 5.94 |
| C | 79 | 3.79 |
| D | 326 | 15.62 |
| E | 1557 | 74.60 |

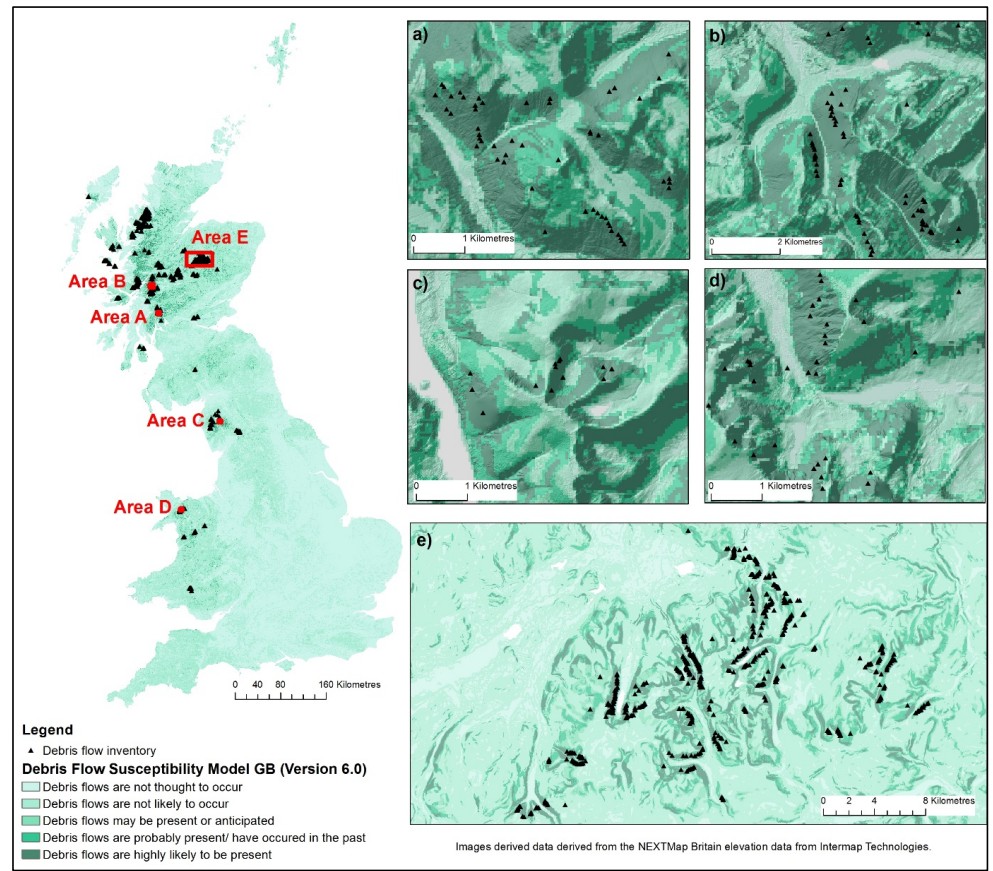

**Figure 9: The debris flow susceptibility model for Great Britain (V6.0). a): Rest and Be Thankful area, Scotland. b):**
**Glen Coe, Scotland. c): Lake District, England. d): Snowdonia, Wales. e): validation area in the Cairngorm Mountains,**
**Scotland.**
To better assess the model performance, a representative area for debris flow occurrence in the Cairngorm
Mountains area (817 km$^2$, with 33 % of the total number of mapped debris flows), was selected for the calculation
of the frequency ratio plot and Receiver Operating Characteristics (ROC) curve. The results are illustrated in Fig.
10. The model satisfies two decision rules considered by Can et al. (2005): (1) most of the observed landslides are
found in the high-susceptibility class, and (2) the high susceptibility class covers as small an area as possible in





the prepared susceptibility map (Table 5). Compared with the overall results, the percentage of debris flows
mapped in classes A and B decreases to 2.89 %, while the percentage of debris flows that occurred within
categories D and E (i.e. areas with the highest potential for instability) increases to 93.50 %.

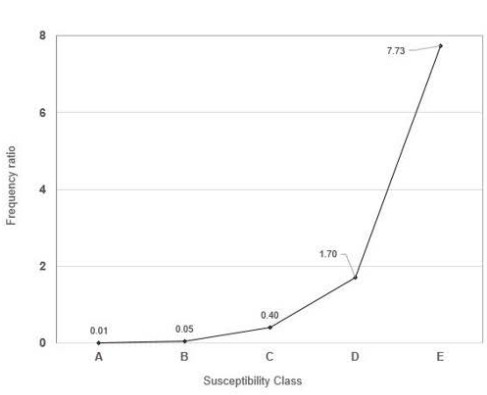
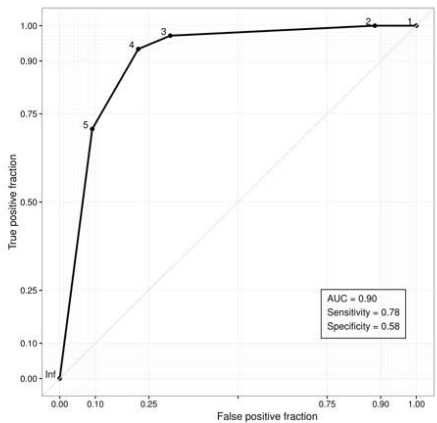

**Figure 10: Model validation using frequency ratio plot (left) and ROC curve (right).**
**Table 5: Statistics of susceptibility classes (total number of debris flows = 691; validation area of = 871 km$^2$).**

| Susceptibility class | No. pixels | % of pixels | No. of debris flows | % of debris flows | Frequency ratio |
|---|---|---|---|---|---|
| A | 37917 | 11.60 | 1 | 0.14 | 0.01 |
| B | 187170 | 57.25 | 19 | 2.75 | 0.05 |
| C | 29363 | 8.98 | 25 | 3.62 | 0.40 |
| D | 42285 | 12.93 | 152 | 22.00 | 1.70 |
| E | 30225 | 9.24 | 494 | 71.49 | 7.73 |

The ROC curve is a tool frequently used in statistical approaches to indicate the general reliability of a
geographical prediction map (Chung and Fabbri 2003, Begueria 2006). Although it cannot reveal the actual
uncertainty of spatial prediction patterns (Vakhshoori and Zare, 2017), it provides a good estimate of the model
accuracy and a common base of comparison between models. The area under the curve (AUC) value of the ROC
curve is 0.90 or 90 % (Fig. 10).
Overall, the proposed susceptibility model seems to perform reasonably well, considering the limitations of
the input data and methodological approach. Those areas where debris flows have been observed seem to
correspond well to areas of high susceptibility highlighted by the model. Given the selection of factors and
attributed scoring values derived through expert elicitation, this new model is conservative but less so and more
accurate in parts of England and Wales than the one developed by Harrison et al. (2006).
There are, of course, limitations to the proposed approach, for example with the accuracy and completeness
of the inventory being used for validation. One challenge when mapping initiation areas using points is the
mismatch between the scale of the process and the model's spatial resolution. As a result, areas adjacent to the



mapped initiation point are not taken into account for model validation. One solution to overcome this problem is
to map the initiation area using polygons.
Another source of errors stemming from the input data is the presence of artefacts in the DTM model (Fig. 7).
It is recognised that the NEXTMap$^{TM}$ elevation model does not always accurately represent the ground surface
and produces erroneous elevation data in given locations. This occurs because of the oblique way in which
NEXTMap$^{TM}$ data are collected. Examples of this include the coast, verges of dense stands of trees and large
structures such as warehouses or extensive stretches of seawall. As a result, debris flow susceptibility values are
therefore likely to be overestimated in these areas. In addition to the artefacts, the spatial resolution of the DTM
was resampled to a coarser resolution (from 5 m to 50 m) to ensure consistency between spatial datasets. For this
reason, the model is not able to reproduce the detailed morphology that could potentially result in a more accurate
model. However, a finer resolution national scale DTM is expected to be available to BGS in the near future and
its effect on the current modelling results will be assessed.
Although heuristic methods introduce uncertainty in model parameters and outputs, similar approaches were
used at the national scale in other study areas and they suggested that, for the most part, the results were
reproducible. For this study, the heuristic approach was deemed to be the most appropriate for the scale of analysis,
data availability and efficient use of peer reviewed studies, expert geologists and geomorphologists at BGS. Such
models offer useful insights to national infrastructure companies when prioritising remediation work to increase
infrastructure resilience from the threat of such hazards.
**4 Conclusions**
The debris flow susceptibility model for Great Britain is a 50 m raster based GIS dataset which provides
information on the potential of the ground, at a given location, to form a debris flow based on a five point scale
from A (debris flows are not thought to occur) to E (debris flows are highly likely to be present). The Model used
expert judgement to combine and weight the relevance of four input factors i.e. the properties and characteristics
of geological materials, slope, influence of stream channels and drainage as the indicators of susceptibility. Those
areas where debris flows have been observed (and recorded in the debris flow inventory) correspond well to areas
of high susceptibility highlighted by the model, with 74 % of recorded debris flows occurring within a category
D or E pixel within the model. The validation results showed that the debris flow susceptibility map satisfies the
decision rules proposed by Can et al. (2005). The ROC curve and frequency plot results support the idea that the
model discriminates reasonably well between areas with potential for landsliding (AUC of 0.90 or a prediction
rate equal to 90 %).
Although not without some limitations, the debris flow susceptibility model for Great Britain has built on
knowledge by Harrison et al. (2006), refining the model and extending its coverage. As such it represents a useful
tool for policy-makers, developers and engineers, and can support regional or national scale development action
plans and disaster risk reduction strategies at the national scale.





**Data availability**
The Debris Flow Susceptibility Model for Great Britain (BGS 2017) and the geological data used to produce this
dataset are available under licence from the British Geological Survey. Please contact enquiries@bgs.ac.uk for
further information.
**Author contributions**
EB was responsible for leading the debris flow product development, including funding acquisition,
conceptualization, investigation, data curation, formal analysis, methodology development and providing
technical expertise to develop the product in GIS. She also the lead in writing this manuscript. CD and CP were
responsible for conceptualization, investigation, formal analysis, methodology development and providing
scientific expertise to underpin development of the product in GIS. They also helped write the manuscript. RC
was responsible for statistical validation of the product, reviewing the methodology and reviewing and editing the
manuscript, including writing original text within the validation section. KL supported the research in a
supervisory capacity, providing consultative geological expertise during model development. All authors
discussed the results and contributed to the final manuscript.
**Competing interests**
The authors declare that they have no conflicts of interest.
**Acknowledgements**
This research was supported by the British Geological Survey's National Capability (NC) funding stream received
from the Natural Environment Research Council (NERC) through UK Research and Innovation (UKRI). The
authors publish this article with the permission of the Executive Director, British Geological Survey (BGS).  We
would like to acknowledge the support of a number of colleagues in the production and editing of this paper and
its underpinning research, namely, Katy Freeborough ,Katy Mee, Catherine Cripps, Russell Lawley, David
Entwistle, Vanessa Banks, Geraldine Wildman, Katherine Royse and Helen Reeves. Whilst every effort has been
made to ensure the accuracy of the material in this paper, the authors will not be liable for any loss or damages
incurred through the use of this paper.

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

Methods for debris flow hazard and risk assessment. Mountain risks: from prediction to management and
governance, 133-177. 2013.
Maccaferri. Debris Flow Barriers - A83 Trunk Road (pt1) (Case History). UK/CH/EP040, available at:
https://www.maccaferri.com/uk/download/ch-rf-uk-debris-flow-barriers-a83-trunk-road-
part3scotland/?wpdmdl=4963. Last access: 01 March 2019. 2014.
Mcmillan, P., Brown, D. J., Forster, A. and Winter, M.G. Debris flow information sources. In: Scottish Road
Network Landslides Study (Eds: Winter, M. G, Macgregor, F. and Shackman, L.). Scottish Executive, pp 25-
25      44. 2005.

Mee, K., Bee, E. J. and Lee, K. User guide and methodology for the Geological Controls on Infiltration GB
(Version 7) GIS dataset. British Geological Survey. IR/16/032. 2016.
Milne, F. D. Topographic material controls of the Scottish debris flow hazard. Doctoral Thesis. University of
Dundee. 2008.
Milne, F. D. Werritty, A., Davies, M. C. R. and Brown, M. J. A recent debris flow event and implications for
hazard management. Quarterly Journal of Engineering Geology and Hydrogeology, 42(1), 51-60. 2009.
Nettleton, I. M., Martin, S., Hencher, S. and  Moore, R.  Debris flow types and mechanisms, in: M. G. Winter, F.
MacGregor & L. Shackman (eds) Scottish Road Network Landslides Study. 2005.
Postance B.F., Hilier, J.K., Dijkstra, T., and Dixon, N. Extending natural hazard impacts: an assessment of
landslide disruptions on a national road transportation network, Environmental Research Letters, 12(1), 1 - 11.
36      2017.

Skinner, K. D. Post-fire debris flow hazard assessment of the area burned by the 2013 Beaver Creek fire near
Hailey, Central Idaho. 2013.
Vakhshoori, V. and Zare, M. Is the ROC curve a reliable tool to compare the validity of landslide susceptibility
maps?, Geomatics, Natural Hazards and Risk, 9(1), 249 – 266. 2018.





Van Westen, C.J. Landslide hazard and risk zonation - why is it still so difficult, Bulletin of Engineering Geology
and the Environment, 65, 167 – 184. 2006.
Winter, M. G., Heald, A.P., Parsons, J.A., Shackman, L. and Macgregor, F.  Scottish debris flow events of August
2004. Quarterly Journal of Engineering Geology and Hydrogeology, 39, 73-78. 2006.
Winter, M. G., Kinnear, N., Shearer, B., Lloyd, L. and Helman, S. A technical and perceptual evaluation of wig-
wag signs at the A83 Rest and be Thankful. Transport Scotland, PPR664. 2013.
Winter, M. G., Macgregor, F. and Shackman, L. Scottish Road Network Landslides Study. The Scottish
Executive. 2005.
Winter, M. G. and Shearer, B. An extended and updated technical evaluation of wig-wag signs at the A83 Rest
And Be Thankful. Transport Research Laboratory. 2017.
Winter, M. G., Smith, J.T., Fotopoulou, S., Pitilakis, K., Mavrouli, O., Corominas, J. and Aegyroudis, S. An expert
judgement approach to determining the physical vulnerability of roads to debris flow. Bulletin of Engineering
Geology and the Environment, 73(2), 291-305. 2014.