# Peer review of "Creating a national scale debris flow susceptibility model for Great Britain"

_Natural Hazards and Earth System Sciences, 2019_

## Referee Comment (RC1) · Anonymous Referee #1 · 20 May 2019

1) Pag. 4 line 13: some GIS based statistical landslide susceptibility assessment have been performed distinguishing between landslide type (see for example Trigila A., Frattini P., Casagli N., Catani F., Crosta G., Esposito C., Iadanza C., Lagomarsino D., Lari S., Scarascia-Mugnozza G., Segoni S., Spizzichino D., Tofani V., 2013 Landslide susceptibility mapping at national scale: the Italian case study. In: K. Sassa, P. Canuti, C. Margottini (eds) Landslide science and practice Vol. 1 Inventory and hazard assessment. Springer, pp. 287-296). 2) Par. 2.2 Predisposing morphological and geological factors: please specify the map scale of the input data (eg. Soil-Parent Material database, GCI, Quaternary Domain Map) 3) Figure 4.d: please check the legend 4) Par. 2.2.3, Table 2: In the text "Scores assigned to slopes overlaying a stream channel

were increased by a factor of two in the 20 - 30° and 15 - 20° categories"; in the table score assigned to the categories are 4 and 6; in the caption "Score increased to 16 or 28". Please check 5) Par. 2.2.3, line 18: "archyrdo tool" instead of "archydro tool" 6) Par. 2.23: I suggest to calculate the distribution of slope angle (histogram) in the study area and at the debris flow point (see for example Van Westen, C.J., Castellanos E., Kuriakose S.L. (2008) Spatial data for landslide susceptibility, hazards and vulnerability assessment: an overview. Engineering geology, 102 (3-4), 112-131) 7) Par. 3 Results and discussion, pag. 14, line 12: please enter "frequency ratio" formula 8) Par. 3, pag. 15, line 12: In the text "Vakhshoori and Zare, 2017"; in References "Vakhshoori and Zare, 2018"; please check 9) Par. 3, pag. 16, line 15-18: in order to consider the heuristic model as the most appropriate on the study area, I suggest to compare it with a statistical model (bivariate/multivariate; see for example Lee S., Min K., 2001 Statistical analysis of landslide susceptibility at Yongin, Korea. Environmental Geology, 40, 1095–1113; Reichenbach P., Rossi M., Malamud B.D., Mihirb M., Guzzetti F., 2018 A review of statistically-based landslide susceptibility models. Earth-Science Reviews 180 (2018) 60–91).

---

## Referee Comment (RC2) · Martina Böhme (Referee) · 6 Jun 2019

This manuscript is presenting the procedure for a national debris flow susceptibility map. The susceptibility model is based on three factors that are weighted based on expert judgement. The manuscript is very well structured, written, and illustrated. However, there are several points that need improvement.

Care should be taken when using the terms susceptibility and hazard. A hazard assessment is something completely different than a susceptibility analysis and cannot be used synonymous.

Different models and different weightings should have been tested and evaluated in order to find the best susceptibility model. Furthermore, the usage of only one parametrisation for the entire country is obviously not appropriate. This is confirmed by the fact that the susceptibility map performs much better in the validation area, then it does for the entire country. Parameters and weights are likely chosen optimal for this validation area, but not for other areas of the country. Topography, geomorphology, geology and climate is varying significantly over different regions in Great Britain and the model should be adapted to this variation. This could be done by separating the model into zones of different regions and landscapes. In addition, the two used parameters "debris material" and "permeability score" are not independent. This may lead to an overestimation of susceptibility in certain areas.

One of my mayor concerns is the use of a 50 m resolution DEM. I think this is too coarse in order to be able to get relevant information regarding the detection of starting zones for debris flows.

I also think, that areas below a certain slope angle should rather be excluded from the model. This is a physical restriction which is independent of all the other parameters.

Several specified data sets are used, but not presented in detail. The manuscript gives no insight about what they present, on what they are based on and what their scales are. In order to evaluate the susceptibility model, detailed information and understanding of the input data is inevitable. I propose rather to use the original data and then implement the algorithms used to produce the specialized data sets directly in the susceptibility model. This would also help to avoid doubled parameters.

Adding a table with all recorded events including the scores they get for each single parameter as well as the final susceptibility score would be beneficial.

More specific comments can be found in the attached commented version of the manuscript.

Please also note the supplement to this comment:
https://www.nat-hazards-earth-syst-sci-discuss.net/nhess-2019-54/nhess-2019-54-RC2-supplement.pdf
* * *
Interactive
comment

[Figure]

**Supplement:**

[revised manuscript text omitted]

---

## Author Comment (AC1) · 31 Jul 2019

Manuscript under review:

**Creating a national scale debris flow susceptibility model for Great Britain: a GIS-based heuristic approach (Emma Bee et al.)**

On behalf of the authors, I am pleased to submit our response to the anonymous Referee's Comments (RC). We appreciate the time invested in the thorough review of this work and have modified the manuscript based on the suggested comments to the best of our abilities.

Below, I have responded to each suggestion or comment following the indicated structure: 1) Referee's comment (**in bold**), (2) author's response (in blue) and (3) author's changes in manuscript.

We believe that the manuscript benefited considerably from your review and look forward to hearing about the results of the evaluation.

Thank you for your consideration

Sincerely,

Emma Bee, Claire Dashwood, Catherine Pennington and Roxana Ciurean

**Referee Comment 1:**
**1) Pg. 4, line 13: Some GIS based statistical landslide susceptibility assessment have been performed distinguishing between landslide type (see for example Trigila A., Frattini P., Casagli N., Catani F., Crosta G., Esposito C., Ladanza C., Lagomarsino D., Lari S., Scarascia-Mugnozza G., Segoni S., Spizzichino D., Tofani V., 2013 Landslide susceptibility mapping at national scale: the Italian case study. In: K. Sassa, P. Canuti, C. Margottini (eds) Landslide science and practice Vol. 1 Inventory and hazard assessment. Springer, pp. 287-296).**

2) Author's response:
Thank you for the reference which we have considered. However, in response to Martina Böhme referee comment 7, in which it was suggested that this section was very long and contains details not relevant to the manuscript, we have shortened this section and rewritten the introduction as described below.

3) Author's change in manuscript:
The term 'debris flow' refers to the rapid downslope flow of poorly sorted debris mixed with water (Ballantyne 2004). Hungr et al. (2014) describe debris flows as "very rapid to extremely rapid surging flows of saturated debris in a steep channel"; widespread in mountainous terrain they are characterised by "strong entrainment of material and water from the flow path". Debris flow initiation can be through a number of mechanisms including shallow landsliding, channel bed mobilisation due to surface water flow, rock fall from a steep bank, seismic activity and are commonly linked to rainfall (Anderson and Sitar, 1995; Hungr et al., 2014; Iverson et al., 1997; Berti and Simoni, 2005; Gabet and Mudd, 2006; Yin et al., 2009).

Debris flows in Great Britain are most commonly found in upland Scotland but also in parts of Wales and the Lake District in England. They occur most commonly following a period of high magnitude precipitation and/or extreme antecedent conditions. According to Nettleton et al. (2005), Ballantyne

(2004) and Cruden (1996) there are two types of debris flow in Great Britain: (a) hillslope or open slope debris flows which form their own path down the valley slopes as tracks or sheets and (b) valley-confined or channelised debris flows which originate in bedrock gullies and are confined for at least part of their length along the gully floor (Fig. 2a and b).

[Figure]

Figure 2: Hillslope (a) and channelised (b) debris flow (Image source: Nettleton et al. 2005)

In Great Britain, the Scottish road and rail networks are recurrently affected by debris flows. In August 2004, two debris flows intersected the A85 in Glen Ogle, north of Lochearnhead, Stirlingshire. Fifty seven people were stranded on the roadway between two debris flows with a cumulative volume of approximately 15000 m3 (Winter et al., 2014) and either left the scene on foot or were rescued by helicopter (Milne et al. 2009). The A85, which normally carries up to 5600 vehicles per day, was closed for four days (Winter et al. 2006). The most widely reported location in Great Britain for debris flow impact on a strategic road is the A83 'Rest and Be Thankful' Pass (British Geological Survey, 2009). Whilst event magnitudes here are generally small, ranging between 200 and 1000 m3 in volume, debris flows have occurred at least on an annual basis over the last 25-30 years (Winter et al. 2014). The road is repeatedly closed in both directions resulting in an 88 km diversion with significant regional economic impact that is regularly reported in the media. Postance et al. (2017) calculated that historic estimates of the economic impact of the 2007 A83 'Rest and Be Thankful' debris flow event totalled £1.2 million over a 15 day closure, 60% greater than previous estimates. Despite the regularity of debris flows on some key strategic routes in Scotland there are no recorded debris flow deaths recorded inland in Great Britain. Recent work by Wong and Winter (2018) suggests the annual probability of a fatality at the A83 Rest and Be Thankful stretch was equivalent to 1 in every 655 years when current mitigation measure are taken into account.

In Great Britain previous studies have concentrated on debris flow susceptibility modelling and hazard ranking in Scotland only. The first regional debris flow susceptibility assessment was the Scottish Road Network Landslides Study (Winter et al., 2005). This was commissioned by the Scottish Executive and conducted by a multidisciplinary working group in response to debris flow events in 2004 that impacted Scotland's road network substantially. The assessment was calibrated by the working group and then interpreted to derive hazard and hazard-ranking information for the Scottish road network (Winter et al., 2013). Areas of England and Wales also prone to debris flows, such as the Lake District and Snowdonia National Park, were excluded from this assessment. To respond to the needs of national asset managers, decision-makers and practitioners, a methodology was developed to create a high level debris flow susceptibility map that displayed the distribution of likely source areas spatially at national scale. The aim of this study was to identify potential debris flow initiation areas and serve as an indication where further, more detailed regional studies should be carried out. Refining the model with a more detailed DTM and further relevant factors such as slope curvature and upslope

contributing area coupled with runout modelling would enable infrastructure, such as roads and railways, in high susceptibility areas to be identified with greater spatial resolution.

**Referee Comment 2:**
**1) Pg.6, Par. 2.2 Predisposing morphological and geological factors: please specify the map scale of the input data (eg. Soil-Parent Material database, GCI, Quaternary Domain Map)**

2) Author's response:
The scale of the indicated maps has now been added in the text.

3) Author's change in manuscript:
In order to create a spatial data layer that classifies geology based on its susceptibility to debris flow occurrence, the BGS 1:50,000 scale Soil–Parent Material Database (Lawley et al., 2009) was analysed to determine the character and availability of regolith and to score the material according to its propensity to fail as a debris flow.

The GCI dataset (fig 5) utilises the BGS 1:50,000 scale permeability datasets for superficial and bedrock geology (Lewis et al. 2006

In order to capture this spatial extent of glacial scouring and the subsequent reduced  likelihood of debris flows in affected areas of North West Scotland the 'ice-scoured montane' domain  of the BGS 1:625,000 scale Quaternary Domain Map (Booth et al., 2015) was utilised

**Referee Comment 3:**
**1) Pg.7, Figure 4.d: please check the legend**

2) Author's response:
The legend for Figure 4d has been corrected.

3) Author's change in manuscript:

[Figure]

**Referee Comment 4:**
**1) Pg.9, Par. 2.2.3, Table 2: In the text "Scores assigned to slopes overlaying a stream channel were increased by a factor of two in the 20 - 30_ and 15 - 20_ categories"; in the table score assigned to the categories are 4 and 6; in the caption "Score increased to 16 or 28". Please check.**

2) Author's response:
The text and table headings have been modified to reflect a clearer description of the methodology.

3) Author's change in manuscript:

In-text: Where slopes with angles of 15°- 20° or 20°- 30° coincided with a stream channel, the score was increased to reflect the observations of Innes (1983) and Milne (2008) that channelised debris flows can initiate on lower angle slopes and to denote their higher potential for debris flow initiation than for open hillslopes with equal gradients.

**Table 2: Scores assigned to slope angle. Score increased to 6 and 8 (from 4 and 6, respectively) only for slopes with angles of 15°- 20° and 20°- 30° coinciding with a stream channel**

| Slope angle (°) | 0-3 | 3-15 | 15-20 | 20-30 | 30-32 | 32-42 | 43-46 | 46-50 | >50 |
|---|---|---|---|---|---|---|---|---|---|

| Score | 0 | 2 | 4 or 6 | 6 or 8 | 8 | 10 | 8 | 6 | 2 |
|---|---|---|---|---|---|---|---|---|---|

**Referee Comment 5:**
**1) RC: Par. 2.2.3, line 18: "archyrdo tool" instead of "archydro tool"**

2) Author's response:
Typo was corrected in the text.

3) Author's change in manuscript:
archydro

**Referee Comment 6:**
**1) Pg.9, Par. 2.23: I suggest to calculate the distribution of slope angle (histogram) in the study area and at the debris flow point (see for example Van Westen, C.J., Castellanos E., Kuriakose S.L. (2008) Spatial data for landslide susceptibility, hazards and vulnerability assessment: an overview. Engineering geology, 102 (3-4), 112-131)**

2) Author's response:
Thank you for the suggestion. Representative subsamples of the study area were selected to analyse the debris flow – slope class relationship. For consistency and ease of comparison, these were the same areas indicated in the manuscript in Figure 9 (including the validation site area).

3) Author's change in manuscript:

[Figure]

Fig. 11 Relationship between slope angle classes and debris flows. Orange line indicates the percentage frequency of debris flows in a given class. Histogram illustrates the distribution of slope angle classes in a) Rest and Be Thankful area (A = 66 km$^2$; number of debris flows, N = 67); b) Glen Coe,

Scotland (A = 45 km$^2$; N = 76); c) Lake District, England (A = 536 km$^2$; N = 65); d) Snowdonia, Wales (A = 88 km$^2$; N = 51); e) Cairngorm Mt., Scotland (A = 817 km$^2$; N = 691). See also Fig. 9.

Figure 11 illustrates the relationship between slope angle and debris flows. In all indicated areas, most debris flows initiations within the inventory are observed on slopes varying between 30° and 40°, in line with the findings of Ballantyne (2004).

**Referee Comment 7:**
**1) Pg.14, Par. 3, line 14: Results and discussion: please enter "frequency ratio" formula**

2) Author's response:
Changes have been made in the text and the formula for frequency ratio has been added.

3) Author's change in manuscript:
The frequency of landslides in a desired class ($FR_i$) is computed as the ratio between the frequency of landslides in the $F_i$ area ($PL_i$) and the frequency of the $F_i$ area ($PF_i$).

$$FR_i = PL_i/PF_i \hspace{3cm} \textbf{(Eq. 2)}$$

where $PL_i$ is the ratio between the area of landslides in the $F_i$ area and the area of landslides in the study area; and $PF_i$ is the ratio between the area of the $F_i$ area to the entire study area (Li et al., 2017). A frequency ratio $FR_i$ larger than 1 indicates that the $i$th class of factor F ($F_i$) favours the occurrence of landslides while the opposite is indicated for $FR_i$ smaller than 1.

**Referee Comment 8:**
**1) Pg.15, Par. 3, line 12: In the text "Vakhshoori and Zare, 2017"; in References "Vakhshoori and Zare, 2018"; please check**

2) Author's response:
The date in the text was corrected.

3) Author's change in manuscript:
Although it cannot reveal the actual uncertainty of spatial prediction patterns (Vakhshoori and Zare, 2018), it provides a good estimate of the model accuracy and a common base of comparison between models.

**Referee Comment 9:**
**1) Pg.16, Par. 3, line 15-18: in order to consider the heuristic model as the most appropriate on the study area, I suggest to compare it with a statistical model (bivariate/multivariate; see for example Lee S., Min K., 2001 Statistical analysis of landslide susceptibility at Yongin, Korea. Environmental Geology, 40, 1095–1113; Reichenbach P., Rossi M., Malamud B.D., Mihirb M., Guzzetti F., 2018 A review of statistically-based landslide susceptibility models. Earth-Science Reviews 180 (2018) 60–91).**

2) Author's response:
A section has been added in the text to detail why the proposed approach was the most appropriate one given the available input data and lack of a specific debris flow inventory.

3) Author's change in manuscript:

At the start of this study a separate inventory for debris flows was not available and so statistical methods of susceptibility mapping were not possible. A national high resolution DTM was not available to allow for a physically based model and so a qualitative heuristic method was used to take advantage of the available data and the broad range of existing studies that had already been undertaken.

---

## Author Comment (AC2) · 31 Jul 2019

Manuscript under review:

**Creating a national scale debris flow susceptibility model for Great Britain: a GIS-based heuristic approach (Emma Bee et al.)**

On behalf of the authors, I am pleased to submit our response to the Referee's Comments (RC). We appreciate the time invested in the thorough review of this work and have modified the manuscript based on the suggested comments to the best of our abilities.

Below, I have responded to each suggestion or comment following the indicated structure: 1) Referee's comment (**in bold**), (2) author's response (in blue) and (3) author's changes in manuscript.

We believe that the manuscript benefited considerably from your review and look forward to hearing about the results of the evaluation.

Thank you for your consideration!

Sincerely,

Emma Bee, Claire Dashwood, Catherine Pennington and Roxana Ciurean

**Referee Comment 1:**
**1) Care should be taken when using the terms susceptibility and hazard. A hazard assessment is something completely different than a susceptibility analysis and cannot be used synonymous.**

2) Author's response:
Thank you for the scrutiny. This comment has been addressed and text altered where appropriate. In addressing other comments regarding shortening sections, some of the text where these terms have been used have been removed from the manuscript.With respect to the paragraph describing the work of Winter et al. (2005; 2013) (lines 23 – 30), we kept the terminology as it is used in the original publication. Herein, the term "hazard" it is used in the widest sense as defined by UNISDR ("A dangerous phenomenon, substance, human activity or condition that may cause loss of life, injury or other health impacts, property damage, loss of livelihoods and services, social and economic disruption, or environmental damage", ISDR, 2009. UN/ISDR Terminology on Disaster Risk Reduction. Geneva, Switzerland, 34 p. http://www.unisdr.org/files/7817_UNISDRTerminologyEnglish.pdf).

3) Author's change in manuscript:
Line 5, page 4: Debris flow susceptibility modelling approaches vary widely depending on the purpose of the analysis, the extent of the study area and available datasets […].

Line 21, page 5: […] identified five main factors to be considered when determining the potential for debris flow initiation […]

**Referee Comment 2:**
**1) Different models and different weightings should have been tested and evaluated in order to find the best susceptibility model. Furthermore, the usage of only one parametrisation for the entire country is obviously not appropriate. This is confirmed by the fact that the susceptibility map performs much better in the validation area, then it does for the entire country. Parameters and**

**weights are likely chosen optimal for this validation area, but not for other areas of the country. Topography, geomorphology, geology and climate is varying significantly over different regions in Great Britain and the model should be adapted to this variation. This could be done by separating the model into zones of different regions and landscapes. In addition, the two used parameters "debris material" and "permeability score" are not independent. This may lead to an overestimation of susceptibility in certain areas.**

2) Author's response:
We have broken down our response into sections to answer each point raised:

**Different models and weighting:** Due to the fact that a debris flow inventory was not available at the onset of this research and the inventory created a-posteriori is not complete, data-driven methodologies were not considered appropriate for the current use. Approaches such as bivariate statistics or logistic regression could not be applied without a more robust inventory. The previous debris flow assessment created by BGS and TRL (Winter et al., 2005) used a heuristic approach which is in line with the current landslide susceptibility map for GB covering rotational/translational landslides. Applying a similar approach allows us to explore the potential for creating a national landslide susceptibility map that combines different assessments (i.e. for debris flows, rock falls, rotational/translational landslides) created with comparable methods.

**Validation:** Parameters and weights were selected to reflect the complexity and heterogeneity of the entire study area. We developed generic parameters and weightings specific to debris flows from an extensive search of the literature covering all major regions in Great Britain. Previous studies also illustrate the difference in geology and topography between these regions indirectly captured through weightings. The nature of debris flows and the geology/topography of GB meant that the distribution of debris flows is restricted to areas of upland terrain with the appropriate geological conditions. Future regional scale assessments will consider various parametrisations specific to each region.

**Parametrisation:** Ballantyne (2004) attributes the spatial trends of debris flows in Scotland predominantly to slope gradient as well as sediment availability and sediment texture. We considered that slope and sediment availability (which takes into account texture) are representative parameters for the entire country. The strong emphasis on parent material (reflecting underlying geological characteristics), slope angle and their respective weights provided a way for adapting the model to different terrain characteristics (e.g., clay rich lithologies or low declivity area were excluded as potential areas for debris flow initiation through weighting).

Landslide domains, which reflect the topography, geomorphology, geology and resultant landslides have been addressed in previous studies (Dashwood *et al*., 2017). These domains will be used to mask those areas where debris flows do not occur in a separate study.

**Independent variables:** Availability of debris material and permeability are not completely independent parameters. However, in this work one reflects the surface conditions whilst the other reflects the underlying conditions impacting on infiltration (see response to Comment 10 for a detailed explanation). Blahut et al. (2010) recognise that many parameters used in landslide susceptibility modelling have some degree of dependence - slope and geology, landuse and slope.

Blahut, J., van Westen, C.J. and Sterlacchini, S. 2010. Analysis of landslide inventories for accurate prediction of debris-flow source areas. Geomorphology Volume 119, Issues 1–2, 15 June 2010, Pages 36-51

3) Author's change in manuscript:

None

**Referee Comment 3:**

**1) One of my major concerns is the use of a 50 m resolution DEM. I think this is too coarse in order to be able to get relevant information regarding the detection of starting zones for debris flows.**

2) Author's response:
The NEXTMap data models of Great Britain are high resolution models of the surface of the earth. The initial product shows the surface of Great Britain down to 5m grid spacing. Processing this data at a national scale is not insignificant. The 5m NEXTMap data was sub-sampled to 50m so that it could be used in visualisation systems and models developed at a national scale, such as this debris flow model. The licensed terms of use for the NextMap product comply with BGS' business model and, at the time of the research, was the only available product we were permitted to use. The 50m subsampled DTM was regarded to be appropriate for the debris flow model because it is a product designed for national scale, for national and regional decision making. In addition, it fitted the mapping scale of other datasets used in this model. BGS has acquired a new DTM. Once this DTM becomes available, we will evaluate its use for national scale studies. However, this will be part of a version 2 debris flow model, not version 1 as outlined in this manuscript.

3) Author's change in manuscript:
Clarified initial scale of DTM capture by adding: The NEXTMap™ 5m Digital Terrain Model (DTM), subsampled to 50m grid cells for a national scale product, was used to obtain the slope angle dataset and derive a stream channel network from flow direction and flow accumulation grids created using the archydro tool in ESRI's ArcGIS. The resolution of the DTM was appropriate for a national scale product and aligned to the other input datasets.

**Referee Comment 4:**

**1) I also think, that areas below a certain slope angle should rather be excluded from the model. This is a physical restriction which is independent of all the other parameters.**

2) Author's response:
A minimum cut off slope angle was discussed during the development of the methodology. There were two reasons we did not adopt one and why we scored lower angle slopes. One was related with the comparability of this dataset with another GeoSure hazard national datasets (rotational/translational landslides) which provides a full coverage of the country. This decision does not affect the overall results, as the proposed methodology allows for the overall score of a low angle slope, even on favourable geology, to be scored low (i.e. unlikely potential for a debris flow to occur). Secondly, a lower angle was used to take into account the potential for debris flows in peat. Strachan et al. (2017; https://www.tandfonline.com/doi/abs/10.1080/14702541.2017.1279679) and Winter et al. (2005) refer to flows initiated in peat on slopes as low as 7° and report that slopes as low as 5° may be used to identify areas with peat. Peat is not systematically mapped across the whole of GB so it was not possible to use a specific slope angle reflecting the distribution peat deposits.

3) Author's change in manuscript:
None

**Referee Comment 5:**

**1) Several specified data sets are used, but not presented in detail. The manuscript gives no insight about what they present, on what they are based on and what their scales are. In order to evaluate the susceptibility model, detailed information and understanding of the input data is inevitable. I propose rather to use the original data and then implement the algorithms used to produce the specialized data sets directly in the susceptibility model. This would also help to avoid doubled parameters.**

2) Author's response:
We have amended the text to ensure the datasets scales are mentioned and described the products developed in other studies as much as possible where this is not published. We have and will also provide references to these studies where literature is available..

3) Author's change in manuscript:

In order to assess the hydrogeological conditions, the BGS developed a national 1:50,000 scale 'Geological Controls on infiltration' (GCI) dataset (Mee et al. 2016) for internal use. The GCI dataset (fig 5) utilises the BGS 1:50,000 scale permeability datasets for superficial and bedrock geology (Lewis et al. 2006, British Geological Survey 2006) and the BGS superficial drift thickness model (Lawley and Garcia-Bajo 2009, British Geological Survey 2009b) to ascertain the ability of water to remain in, or infiltrate through, the geological material forming and underneath the potential debris flow. Following the methodology described by Dearden el al. (2013), the superficial and bedrock permeability datasets were assigned the following scores: 1 for free draining, 2 for highly variable or 3 for poorly draining, based on their relative minimum and maximum permeability. The superficial drift thickness model was then used to identify superficial units above and below a threshold thickness of 3 metres. This threshold was determined through discussions with BGS hydrogeologists and reflects the cut-off at which drainage is likely controlled by the permeability of the superficial layer (> 3 m) or a combination of both superficial and bedrock permeability (< 3 m). Based on this information, an overall score for the GCI was generated as follows: where drainage was assumed to be controlled by the superficial geology alone, the score for the superficial drainage was retained (grey cells in Table 1). In the absence of superficial deposits, the bedrock geology drainage score was retained (white cells in Table 1). In scenarios where the drainage was thought to be controlled by a combination of bedrock and superficial deposits, a matrix was used to derive the final score (green, yellow and red cells in Table 1).

**Table 1. Matrix informing the Geological Controls on Infiltration GB dataset (V7) (Mee et al., 2016) used to derive permeability scores**

| | | | Superficial deposits < 3 m | | |
|---|---|---|---|---|---|
| | | | 3 | 2 | 1 |
| | | | Poorly draining | Highly variable permeability | Free draining |
| **Bedrock** | 1 | Free draining | 1 | 1 | 1 |
| | 2 | Highly variable permeability | 2 | 2 | 3 |
| | 3 | Poorly draining | 3 | 3 | 3 |

These scores were then adjusted to reflect a 10-point scale to align with other input data sets being used within the debris flow model and associated descriptions were generated (Table 2).

**Table 2: Infiltration score within the Geological Controls on Infiltration GB dataset (V7) and their relevance to the potential debris flow occurrence.**

| Score | GCI Description | Relevance to debris flow occurrence | Reclassified score |
|---|---|---|---|
| 1 | The subsurface is likely to be free draining | The infiltration conditions are such that, depending on the presence or not of other determining factors (e.g. slope, characteristics of geological material.) the potential for debris flows is low | 1 |
| 2 | The subsurface is likely to have highly variable drainage | The infiltration conditions are such that, depending on the presence or not of other determining factors (e.g. slope, characteristics of geological material etc.) the potential for debris flows is moderate | 5 |
| 3 | The subsurface is likely to be poorly draining | The infiltration conditions are such that, depending on the presence or not of other determining factors (e.g. slope, characteristics of geological material etc.) the potential for debris flows is high | 10 |

**Referee Comment 6:**
**1) Adding a table with all recorded events including the scores they get for each single parameter as well as the final susceptibility score would be beneficial**.

2) Author's response:
As part of BGS business model, not all data is publicly available without licence. The dataset developed in this study may form part of a commercially available product and as such, currently, we are not able to release it in the way proposed.

3) Author's change in manuscript:
None

SPECIFIC COMMENTS NOTED IN THE MANUSCRIPT (OTHER THAN THOSE ADDRESSED ABOVE)

INTRODUCTION

**Referee Comment 7:**
**1) Pg. 1: This section is very long and contains to many details that are not relevant for this manuscript. Should be shortened.**

2) Author's response:
The Introduction section was shortened and only relevant information was retained.

3) Author's change in manuscript:
1 Introduction

The term 'debris flow' refers to the rapid downslope flow of poorly sorted debris mixed with water (Ballantyne 2004). Hungr et al. (2014) describe debris flows as very rapid to extremely rapid surging flows of saturated debris in a steep channel; widespread in mountainous terrain they are characterised by strong entrainment of material and water from the flow path. Debris flow initiation can be through a number of mechanisms including shallow landsliding, channel bed mobilisation due to surface water flow, rock fall from a steep bank, seismic activity and are commonly linked to rainfall (Anderson and Sitar, 1995; Hungr et al., 2014; Iverson et al., 1997; Berti and Simoni, 2005; Gabet and Mudd, 2006; Yin et al., 2009).

Debris flows in Great Britain are most commonly found in upland Scotland but also in parts of Wales and the Lake District in England. They occur most commonly following a period of high magnitude precipitation and/or extreme antecedent conditions. According to Nettleton et al. (2005), Ballantyne (2004) and Cruden (1996) there are two types of debris flow in Great Britain: (a) hillslope or open slope debris flows which form their own path down the valley slopes as tracks or sheets and (b) valley-confined or channelised debris flows which originate in bedrock gullies and are confined for at least part of their length along the gully floor (Fig. 2a and b).

[Figure]

Figure 2: Hillslope (a) and channelised (b) debris flow (Image source: Nettleton et al. 2005)

In Great Britain, the Scottish road and rail networks are recurrently affected by debris flows. In August 2004, two debris flows intersected the A85 in Glen Ogle, north of Lochearnhead, Stirlingshire. Fifty seven people were stranded on the roadway between two debris flows with a cumulative volume of approximately 15000 m3 (Winter et al., 2014) and either left the scene on foot or were rescued by helicopter (Milne et al. 2009). The A85, which normally carries up to 5600 vehicles per day, was closed for four days (Winter et al. 2006). The most widely reported location in Great Britain for debris flow

impact on a strategic road is the A83 'Rest and Be Thankful' Pass (British Geological Survey, 2009). Whilst event magnitudes here are generally small, ranging between 200 and 1000 m3 in volume, debris flows have occurred at least on an annual basis over the last 25-30 years (Winter et al. 2014). The road is repeatedly closed in both directions resulting in an 88 km diversion with significant regional economic impact that is regularly reported in the media. Postance et al. (2017) calculated that historic estimates of the economic impact of the 2007 A83 'Rest and Be Thankful' debris flow event totalled £1.2 million over a 15 day closure, 60% greater than previous estimates. Despite the regularity of debris flows on some key strategic routes in Scotland there are no recorded debris flow deaths recorded inland in Great Britain. Recent work by Wong and Winter (2018) suggests the annual probability of a fatality at the A83 Rest and Be Thankful stretch was equivalent to 1 in every 655 years when current mitigation measure are taken into account.

In Great Britain previous studies have concentrated on debris flow susceptibility modelling and hazard ranking in Scotland only. The first regional debris flow susceptibility assessment was the Scottish Road Network Landslides Study (Winter et al., 2005). This was commissioned by the Scottish Executive and conducted by a multidisciplinary working group in response to debris flow events in 2004 that impacted Scotland's road network substantially. The assessment was calibrated by the working group and then interpreted to derive hazard and hazard-ranking information for the Scottish road network (Winter et al., 2013). Areas of England and Wales also prone to debris flows, such as the Lake District and Snowdonia National Park, were excluded from this assessment. To respond to the needs of national asset managers, decision-makers and practitioners, a methodology was developed to create a high level debris flow susceptibility map that displayed the distribution of likely source areas spatially at national scale. The aim of this study was to identify potential debris flow initiation areas and serve as an indication where further, more detailed regional studies should be carried out. Refining the model with a more detailed DTM and further relevant factors such as slope curvature and upslope contributing area coupled with runout modelling would enable infrastructure, such as roads and railways, in high susceptibility areas to be identified with greater spatial resolution.

**Referee Comment 8:**
**1) Pg.1: Check formatting of in-text citation.**

2) Author's response:
All in-text citations formatting were checked and amended where needed.

3) Author's change in manuscript:
See manuscript.

**Referee Comment 9:**
**1) Pg. 2: (Levees) should be marked in the figure.**

2) Author's response:
Figure was modified.

3) Author's change in manuscript:

[Figure]

RATIONALE FOR RESEARCH

**Referee Comment 10:**
   1)  Pg. 3: Rationale for research section - very long and contains too any details not relevant to the research esp 2nd and 3rd paragraphs.

2) Author's response:
We have edited this section to reduce its length.

3) Author's change in manuscript:
See response to Comment 7.

**Referee Comment 11:**
**1) Pg. 4, Sentence: "While statistical models for potential source identification are based on extensive inventories of past events (Blahut et al. 2010, Carrara et al. 2008), deterministic approaches consider physical characteristics of the process and are thus transferable to any site (Iovine et al. 2003) but are high data demanding and require calibration, which makes them rarely feasible for national scale applications"**

**RC states - Not necessarily, see for example Horton et al. 2013 (https://www.nat-hazards-earth-syst-sci.net/13/869/2013/nhess-13-869-2013.pd), section 2.1 "Assessment of the source area". This is a very simple deterministic approach. The same procedure was applied on a national scale for entire Norway (https://www.ngu.no/en/topic/susceptibility-maps-debris-slides-and-debris-flows; https://www.ngu.no/upload/Publikasjoner/Rapporter/2014/2014_019.pdf; https://www.ngu.no/upload/Ansattbilder/cv/fischer%20et%20al2012.pdf).**

2) Author's response:
Thank you for the useful references.

Our understanding of the different methods of debris flow modelling aligns with that of Mergilli et al. 2012 (https://link.springer.com/content/pdf/10.1007%2Fs11069-011-9965-7.pdf, page 4), where deterministic models are physically based approaches that use rheological assumptions. These are for

example physically based models populated with geotechnical data such as TRIGRS (Montgomery and Dietrich, 1994). Soeters and Van Westen (1996) recommend the application of deterministic analysis at a scale of 1:10,000 not at a national scale and this is more what we were reflecting in our original text.

The model illustrated through the cited papers (FlowR) is not a deterministic model in the sense presented above because it does not take into account local controlling factors and actual physical behaviours. At best, it can be viewed as a semi-deterministic model due to the use of the Perla et al. (1984) friction model - among other. Horton et al. (2013) consider it an empirical approach because it allows the user to calibrate the model parameters based on inventories of past events.

We are pleased to learn about the application of this model at national scale in Norway and would be interested to know if there is an English translation of the NGU report available. As the study shows, the successful application of FlowR is conditioned by the use of a good quality DEM. For a regional to national scale that would typically be 10 – 25m, not 50 m as it is in our case. Moreover, the calibration and validation of model results require information about the observed inundation areas, which are only presently being produced at BGS. Future work is focusing on application of the FlowR model for regional scale assessments in Scotland.

3) Author's change in manuscript:
Debris flow susceptibility models vary widely depending on the purpose of the analysis, the extent of the study area and available datasets. Complex numerical (Chen and Lee, 2000) and deterministic models (Iovine et al., 2003) consider physical characteristics of the process and its behaviour and are mostly applied at slope scale.  Probabilistic (Chang et al., 2014), statistical (Qin et al., 2019; Carrara et al., 2008) and empirical models (Blahut et al., 2010; Horton et al., 2013) are based on extensive inventories of past events and field observations respectively,  and can be tested on a range of scales from slope to regional.  Modelling of both debris flow source area and propagation requires the collection of site-specific data generally not available over large areas. As such, regional and national scale studies tend to employ less data-demanding models and focus on the prediction of debris flow initiation (Fischer et al., 2012).

At the start of this study a separate inventory for debris flows was not available and so statistical methods of susceptibility mapping were not possible. A national high resolution DTM was not available to allow for a physically based model and so a qualitative heuristic method was used to take advantage of the available data and the broad range of existing studies that had already been undertaken.

DATA AND METHODS

**Referee Comment 12:**
**1) Pg. 5: Some more detail on the input data is needed: resolution, source, ...**

2) Author's response:
This section was restructured and extended to contain more detailed information.

3) Author's change in manuscript:
See response to comments below.

**Referee Comment 13:**
**1) Pg. 5: Some more details about this would be good. Was this done for entire GB? How was it done? What events do you expect to discover with this method (age and size limitations)? Refer to map that show those.**

2) Author's response:
The requested information was added in the text.

3) Author's change in manuscript:
In order to develop a national scale debris flow susceptibility model for Great Britain, the characterisation of the geological and geomorphological factors that increase the likelihood of debris flow occurrence and their potential to be represented spatially within a GIS was explored. Debris flows have a distinct morphology which allows for their identification using remote sensing techniques. Aerial photographs (1:10,000) and LiDAR imagery (2m) were utilised to create an inventory of over 2000 debris flows across selected upland areas in Great Britain. Not all debris flows are detectable through this methodology, some known debris flows have subsequently been revegetated and the features are no longer visible. However, debris flows which are recharged with sediment and reactivated or that are seasonally active could be more readily identified. The aerial photographs had a ground resolution cell size of 25cm and this allows high contrast features of around 2.5m to be identified (Mantovani et al., 1996) which is approximately the width of some of the narrower debris flows which were mapped. Characteristic morphological features used to distinguish debris flow fans from other sediment-laden process depositional areas in upland areas included: high slope angle of the fan, very large individual particles, coarse levées and boulder trains, signs of impact loading on obstacles, U-shaped eroded channels and steep, debris-loaded channels upstream (Hungr et al. 2014). Identifiable features which could be observed included the upper, erosional section of the flow (gully), parallel levées of dominantly coarse debris (Fig. 1) and downslope lobes of bouldery debris (Ballantyne 2004) which were utilised to locate an inventory point at the highest part of the upper erosional section.

**Referee Comment 14:**
**1) Pg. 5: What is the size of this test area compared to the size of the entire study area?**

2) Author's response:
The requested information was added in the text.

3) Author's change in manuscript:
The model was evaluated using a frequency ratio plot and the Receiver Operating Characteristics (ROC) curve in a representative area for debris flow occurrence (817 km$^2$, 0.39% of the study area).

**Referee Comment 15:**
**1) Pg. 5: Your decision should be based on several other studies, not only one (i.e. Carrara et al. 2008). Especially as this study was based in an alpine environment and it is not sure if you can transfer their findings directly to your study area.**

2) Author's response:
The decision of selecting the factors that increase the likelihood of debris flow occurrence was made based on a study performed in Scotland, as indicated in the text (i.e. Winter et al. 2005) which was confirmed by the findings of Carrara et al. (2008).

3) Author's change in manuscript:
None

**Referee Comment 16:**
**1) Pg. 6: Coarse grained material also generally have good permeability thus two parameters debris material and permeability score are not independent**

2) Author's response:
Our model contains two components of lithology/sediment type but with different uses. One reflects the surface conditions (availability of source material) whilst the other reflects the underlying conditions impacting on infiltration. We consider these two parameters are relevant and provide useful information for the landslide susceptibility model.

The two different uses of lithology/parent material are:

**Availability of sediment** - A slope with no sediment available for transport will not be a source for debris flows. So we first had to assess the presence of material that could fail and we used the parent material model for this as it contained information on texture, lithology and what the material weathered to. The scoring system weights a granular superficial deposits as high and a bedrock lithology that is unlikely to weather to a significant regolith as low.

**Hydrogeology/Infiltration** - Secondly, rate of infiltration and permeability of the underlying material is of interest to debris flow initiation. Nettleton et al., 2005 observe that "*certain geological conditions are prone to the effects of water infiltration, for example where permeable soil overlies less permeable bedrock*" The hydrogeology factor takes not just the surface geology into account but the underlying material to try to account for the potential geological conditions mentioned by Nettleton et al., 2005, where differences in permeability mentioned could lead to parallel seepage, perched water tables and impediment of infiltration. We have included the Campbell (1975) diagram below to reflect our attempts to capture not just the permeability of the material available for debris flows to initiate in but also the underlying material which influence the development of perched water tables and downslope seepage forces.

[Figure]

Campbell, 1975. (Campbell, R.H. 1975. Soil slips, debris flows and rainstorms in the Santa Monica Mountains and vicinity, Southern California. U.S. Geological Survey Professional paper 851. 51p)

3) Author's change in manuscript:

See Comment 23 below for a summary of the above.

**Referee Comment 17:**
**1) Pg. 6: This entire part is a direct copy from Lawley et al. (2009). This is not acceptable as it is. Either you mark it as a quotation or reformulate it.**

2) Author's response:
The text was amended as suggested.

3) Author's change in manuscript:
In order to create a spatial data layer that classifies geology based on its susceptibility to debris flow occurrence, the BGS 1:50,000 scale Soil–Parent Material Database (Lawley et al., 2009) was analysed to determine the character and availability of regolith and to score the material according to its propensity to fail as a debris flow. Lawley et al. (2009) describe a 'soil parent material' as "a geological deposit over, and within which, a soil develops".

**Referee Comment 18:**
**1) Pg 6 – Permeability – porosity (drainage) - overlapping with permeability score and may coarse over scoring of some source areas**

2) Author's response:
Please see comment 16.

3) Author's change in manuscript:
None

**Referee Comment 19:**
**1) Pg. 6: GIS based logical decision tree algorithm – present this.**

2) Author's response:
BGS' business model is such that it licences a number of its data products at a fee. Releasing detailed information about the methodology could compromise this business model. However, we agree that the paper would benefit from a decision tree diagram but to protect the IPR, it will not detail the expert derived scoring.

3) Author's change in manuscript:

[Figure]

Figure 4: Logical decision-tree algorithm to determine the propensity of each soil–parent material to fail as a debris flow (scaled from 1 - low susceptibility to 10 - high susceptibility)

**Referee Comment 20:**
**1) Page 6 – include a table with all the materials and their respective scale.**

2) Author's response:
   Refer to response to comment 19 above

3) Author's change in manuscript:
Refer to response to comment 19 above

**Referee Comment 21:**
**1) Pg. 6: Thickness of material being inferred by expert -  How was this done? Was one value assumed per material being valid for the entire country? This is usually very variable. I also do not understand how it was included for the scale? Was there a threshold limit set? Debris flows can originate in very variable sediment thicknesses. Clarify this.**

2) Author's response:
We have attempted to clarify the use of thickness in the model in the manuscript text.

3) Author's change in manuscript:
Thickness was used to distinguish between those deposits that would provide a thicker amount of available sediment for entrainment/failure such as a superficial, unconsolidated gravel and sands compared to an igneous rock. A minimum depth of regolith is quoted as 0.3m in Scotland (Milne et al., 2012) and this was used to as a guide to qualitatively rank the relative thickness scores between deposits. Broad, experience-based assumptions were made, backed by available literature, on the basis that fine grained igneous rocks were less likely to form a sufficient thickness of regolith compared to a course grained sedimentary deposit. Granular superficial deposits such as talus were assumed to have the greatest thickness of available sediment

**Referee Comment 22:**
**1) Pg.6: The difference between superficial and surficial needs to be clarified.**

2) Author's Response:
Changes to text were made.

3) Author's change to manuscript:
The algorithm considered the substrate of the geological material (i.e. whether it was bedrock, superficial or a surficial [a thin veneer] deposit), its origin (i.e. igneous, sedimentary or metamorphic) and the parent-materials strength characteristics when weathered to produce a map showing the propensity for the parent material at any given location, to fail as a debris flow (Fig. 5).

**Referee Comment 23:**
**1) Pg.7: Hydrogeological conditions - What does the first sentence mean for the material. This sentence should be linked up to the rest of the section.**

2) Author's response:

Changes made to the start of the paragraph.

3) Author's change in manuscript:

Empirical evidence from Scotland suggests that many of the debris flows are "*triggered by short intense rainfall preceded by periods of heavy (antecedent) rainfall*" (Winter et al., 2005). The initiation of rainfall-induced debris flows occurs due to the rapid infiltration of water or saturation of the soil mass and a subsequent rise in pore water pressure (Wieczorek, 1996). Pore pressure changes due to snow melt can also trigger debris flows due to a build-up of pore water pressures and this has been identified as the cause of a recent debris flow that derailed a train in January 2018 (RAIB, 2018). Nettleton *et al*. (2005) observe that "*certain geological conditions are prone to the effects of water infiltration, for example where permeable soil overlies less permeable bedrock*". For this reason the hydrogeological component of the model attempts to assess the infiltration potential of the slopes taking into account the permeability of the material underlying the potentially mobile deposit. For instance, a slope that has been glacially scoured is likely to have bedrock cropping out at the surface. This would lower its score for debris flow initiation, especially if the bedrock is a material that does not produce a significant regolith. The glacially scoured bedrock material may also be impermeable, but in this scenario the infiltration/permeability component is less important than the availability of sediment. In a different scenario the presence of a mobile sediment overlying a less permeable bedrock would increase the likelihood of debris flow initiation because of the inability of water to infiltrate the underlying bedrock and the consequent build-up of pore pressures.

**Referee Comment 24:**
**1) Pg. 8: Based on which data is this? I guess the soil type is having an influence here too. You are thus including the soil type twice.**

2) Author's response:
Please see comment 16.

3) Author's change in manuscript:
None

**Referee Comment 25:**
**1) Pg. 8: This entire paragraph is not clear without studying all the references (that are also not accessible login needed). Clearify better what the different datasets are containing.**

2) Author's response:
This section has been amended.

3) Author's change in manuscript:
Empirical evidence from Scotland suggests that many of the debris flows are "triggered by short intense rainfall preceded by periods of heavy (antecedent) rainfall" (Winter et al., 2005). The initiation of rainfall-induced debris flows occurs due to the rapid infiltration of water or saturation of the soil mass and a subsequent rise in pore water pressure (Wieczorek, 1996). Pore pressure changes due to snow melt can also trigger debris flows due to a build-up of pore water pressures and this has been identified as the cause of a recent debris flow that derailed a train in January 2018 (RAIB, 2018). Nettleton et al. (2005) observe that "certain geological conditions are prone to the effects of water infiltration, for example where permeable soil overlies less permeable bedrock".  For this reason the

hydrogeological component of the model attempts to assess the infiltration potential of the slopes taking into account the permeability of the material underlying the potentially mobile deposit. Rapid infiltration of water in a slope formed of soil or colluvial material can lead to the development of perched water tables and downslope seepage forces if the rate of infiltration exceeds the rate of infiltration into the bedrock below (Turner, 1996)In order to assess the hydrogeological conditions, the BGS developed a national 1:50,000 scale 'Geological Controls on infiltration' (GCI) dataset (Mee et al. 2016) for internal use.

The GCI dataset (fig 5) utilises the BGS 1:50,000 scale permeability datasets for superficial and bedrock geology (Lewis et al. 2006, British Geological Survey 2006) and the BGS superficial drift thickness model (Lawley and Garcia-Bajo 2009, British Geological Survey 2009b) to ascertain the ability of water to remain in, or infiltrate through, the geological material forming and underneath the potential debris flow. Following the methodology described by Dearden el al. (2013), the superficial and bedrock permeability datasets were assigned the following scores: 1 for free draining, 2 for highly variable or 3 for poorly draining, based on their relative minimum and maximum permeability. The superficial drift thickness model was then used to identify superficial units above and below a threshold thickness of 3 metres. This threshold was determined through discussions with BGS hydrogeologists and reflects the cut-off at which drainage is likely controlled by the permeability of the superficial layer (> 3 m) or a combination of both superficial and bedrock permeability (< 3 m). Based on this information, an overall score for the GCI was generated as follows: where drainage was assumed to be controlled by the superficial geology alone, the score for the superficial drainage was retained (grey cells in Table 1). In the absence of superficial deposits, the bedrock geology drainage score was retained (white cells in Table 1). In scenarios where the drainage was thought to be controlled by a combination of bedrock and superficial deposits, a matrix was used to derive the final score (green, yellow and red cells in Table 1).

**Referee Comment 26:**
**1) Pg. 8,Table 1: Are these the original GCI descriptions, or did you modify them? How were those scores defined?**

2) Author's Response:
The descriptions were modified from the GCI dataset which attempted to make them relevant for debris flow initation. As such the original matrix which included information on the thickness and permeability of the bedrock and superficial deposits (see below) was reclassified into 3 categories presented in the manuscript. The text was changed to reflect the modification.

3) Author's change in manuscript:

Changes made - See response to comment 5.

**Referee Comment 27:**
**1) Pg.9: Proximity of stream channels - I cannot see how this parameter is included in the susceptibility model. I also suggest to use more parameters (e.g. planar curvature, size of the water catchment area, see Horten et al. 2013).**

2) Author's response:

We have attempted to clarify the text. With respect to the Horton et al. (2013) model, see response to Comment 11.

3) Author's change in manuscript:
Where slopes with angles of 15°- 20° or 20°-30° coincided with a stream channel, the score was increased to reflect the observations of Innes (1983) and Milne (2008) that channelised debris flows can initiate on lower angle slopes and to denote their higher potential for debris flow initiation than for open hillslopes with equal gradients.

**Referee Comment 28:**
**1) Pg. 9: Flow direction, flow accumulation and network - How is this used for the susceptibility model?**

2) Author's response:
Changes were made in the manuscript for more clarification.

3) Author's change to manuscript:
The NEXTMap™ 50 m Digital Terrain Model (DTM), subsampled to 50m grid cells for a national scale product, was used to obtain the slope angle dataset and derive a stream channel network from flow direction and flow accumulation grids created using the archydro tool in ESRI ArcGIS.

**Referee Comment 29:**
**1) Pg. 9, Table 2: There is a misfit with the values in the table and text (factor of two?). What is the meaning of the superscripts?**

2) Author's Response:
The misfit was corrected and text clarified. The superscripts were eliminated.

3) Author's change to manuscript:
Where slopes with angles of 15°- 20° or 20°- 30° coincided with a stream channel, the score was increased to reflect the observations of Innes (1983) and Milne (2008) that channelised debris flows can initiate on lower angle slopes and to denote their higher potential for debris flow initiation than for open hillslopes with equal gradients.

**Table 2: Scores assigned to slope angle. Score increased to 6 and 8 (from 4 and 6, respectively) only for slopes with angles of 15°- 20° and 20°- 30° coinciding with a stream channel**

| Slope angle (°) | 0-3 | 3-15 | 15-20 | 20-30 | 30-32 | 32-42 | 43-46 | 46-50 | >50 |
|---|---|---|---|---|---|---|---|---|---|
| Score | 0 | 2 | 4 or 6 | 6 or 8 | 8 | 10 | 8 | 6 | 2 |

**Referee Comment 30:**
**1) Pg. 9, Table 2: This table might be ok for real slope angles. However, using a 50 m DEM, means a significant smoothing of the topography. Scores have to be adapted to this. Check the distribution of slope angles in the study area and for the recorded events.**

2) Author's response:
These categories were developed with a finer resolution DTM in mind and reflect the slope angles quoted in the literature for GB. We have highlighted this in the text as below.

Representative subsamples of the study area were selected to analyse the debris flow – slope class relationship. For consistency and ease of comparison, these were the same areas indicated in the manuscript in Figure 9 (including the validation site area).

3) Author's change in manuscript:

These categories were developed with a fine resolution DTM in mind, however, a fine resolution DTM was not available or useable at a national scale without considerable processing challenges at the time this research was conducted.

Once a finer resolution DTM becomes available and, the processing challenges overcome, the authors aspire to develop a more regionally specific iteration of the model that could include factors such as curvature and contributing upslope area in combination with run out modelling.

[Figure]

**Fig. 11 Relationship between slope angle classes and debris flows. Orange line indicates the percentage frequency of debris flows in a given class. Histogram illustrates the distribution of slope angle classes in a) Rest and Be Thankful area (A = 66 km²; number of debris flows, N = 67); b) Glen Coe, Scotland (A = 45 km²; N = 76); c) Lake District, England (A = 536 km²; N = 65); d) Snowdonia, Wales (A = 88 km²; N = 51); e) Cairngorm Mt., Scotland (A = 817 km²; N = 691). See also Fig. 9.**

Figure 11 illustrates the relationship between slope angle and debris flows. In all indicated areas, most debris flows initiations within the inventory are observed on slopes varying between 30° and 40°, in line with the findings of Ballantyne (2004).

**Referee Comment 31:**
**1) Pg. 11: BGS Quaternary Domain Map – resolution or scale?**

2) Author's response:

The requested information was added in the text.

3) Author's change in manuscript:
In order to capture this spatial extent of glacial scouring and the subsequent reduced likelihood of debris flows in affected areas of North West Scotland the 'ice-scoured montane' domain of the BGS 1:625,000 scale Quaternary Domain Map (Booth et al., 2015) was utilised..

**Referee Comment 32:**
**1) Pg. 12: There might be less material necessary to initiate a slide in this setting, as the glacially scored surfaces are very smooth, thus having less resistance.**

2) Author's response:
More information with regard to the use of glacial scouring was added in the text.

3) Author's change in manuscript:
Visual comparison of areas with high debris flow potential and the aforementioned debris flow inventory highlighted a number of areas with very limited or no recorded incidence of debris flows in Outer Hebrides, Knoydart, Morven and Argyll. This matched observations of Ballantyne (2004) who attributed this scarcity to extensive glacial scouring (Fig. 9). Areas in the north-west Highlands of Scotland have been subjected to extensive areal scouring (Gordon, 1981) leaving glacially polished bedrock at the surface with little or no regolith or superficial deposits and therefore less material available for debris flows to occur.

In order to capture this spatial extent of glacial scouring and the subsequent reduced likelihood of debris flows in affected areas of North West Scotland the 'ice-scoured montane' domain of the BGS Quaternary Domain Map (Booth et al., 2015) was utilised (Fig. 9a). The ice scoured domain is characterised as "acute variations in relief and extensive, unweathered bedrock outcrop (80%). Soil and weathering products are thin or non-existent, and superficial deposits are patchy in distribution and very variable in type and thickness" (Booth et al., 2015).

**Referee Comment 33:**
**1) Pg. 13: I think the recorded events should be including in this step.**

2) Author's response:
Please see response to Comment 6.

3) Author's change in manuscript:
None

RESULTS AND DISCUSSION

**Referee Comment 34:**
**1) Pg. 13: "However, approximately 6 % of the mapped debris flows were attributed to category A or B, which, according to the model, would have suggested that debris flows are unlikely or not thought to occur" - discuss reasons for this .**

2) Author's response:

We have checked each debris flow within category A and B (6%) and considered the underpinning data at each location. The reasons for having for having debris flows on A's and B's can be summarised as:

1) SPATIAL LOCATION : Less than 1% of the inventory have been incorrectly located at the time of mapping.
2) SLOPES: The resolution of the dtm (50m) means that in some cases, a steep slope near to a ridge feature has been captured as a lower angled slope.
3) CARTOGRAPHIC ACCURACY OF GEOLOGICAL MAPPING: The cartographic accuracy of the geological map (and Parent Material dataset) is 50m on the ground. If a debris flow inventory point occurred on a geological boundary between a low and high scoring geological formation then it may fall within the map in the low scoring area, whereas in reality, given the cartographic error, may fall on the high scoring geological unit.
4) MAPPING OF SUPERFICIAL DEPOSITS - Sometimes superficial deposits aren't mapped consistently, therefore you may have a debris flow occurring that isn't mapped. These are beyond our control, and limitations of the underpinning geological data.

3) Author's change in manuscript:

   The reasons for this are likely to be attributed to one of the following four limitations of the model: 1) the spatial accuracy of the mapped debris flows within the inventory; 2) the resolution of the DTM means that in some cases a steep slope near to a ridge feature may have been captured as a lower angled slope; 3) the cartographic accuracy of the underpinning geological data is 50m on the ground, meaning that if a debris flow inventory point occurred on a geological boundary between a low and high scoring geological formation then it may coincide on the map with the low scoring area, rather than high; 4) inconsistency in mapping of superficial deposits.

**Referee Comment 35:**
**1) Pg. 14: Is it acceptable that a lower susceptibility class has a higher number of recorded events? Maybe some adaptations in the model are needed? Give also the area covered by each class (or number of pixels) as in table 5.**

2) Author's response:
Yes, it is possible that a lower susceptibility class has a higher number of recorded events if, as recognised, the area it covers is larger than that of a higher susceptibility class. The frequency ratio (FR) is a good indicator for comparison. In this case, it shows an increase from class A to E (low to high susceptibility) which is comparable with the FR of the validation area. The table was updated in the manuscript.

3) Author's change in manuscript:
**Table 4: Comparison of debris flow model for Great Britain (v6.0) against mapped debris flow occurrences (n = 2087).**

| Class | No. pixels | % of pixels | No. of debris flows | % of debris flows | Frequency ratio |
|-------|-----------|-------------|---------------------|-------------------|-----------------|
| A | 52543449 | 56.29 | 1 | 0.05 | 0.001 |
| B | 35753967 | 38.30 | 124 | 5.94 | 0.16 |
| C | 2148372 | 2.30 | 79 | 3.79 | 1.64 |
| D | 1696509 | 1.82 | 326 | 15.62 | 8.60 |

| E | 1208859 | 1.29 | 1557 | 74.60 | 57.61 |
|---|---|---|---|---|---|

**Referee Comment 36:**
**1) Pg. 14: I recommend to use a different color scale. This one is hard to differentiate.**

2) Author's response:
BGS corporate style for its 'data products' necessitates using a single colour ramp. The colour scheme used for the debris flow model is in keeping with this house style and has been selected to align to similar datasets BGS produces. We do not propose to make any changes as this would go against our corporate branding.

3) Author's change in manuscript:
None.

**Referee Comment 37:**
**1) Pg. 14: Refer to Fig. 9e. " […] the high susceptibility class covers as small an area as possible […] "
- Check word order.**

2) Author's response:
Text was amended.

3) Author's change in manuscript:
[…] the Cairngorm Mountains area (817 km$^2$, with 33 % of the total number of mapped debris flows; Fig. 10e). […] the high susceptibility class covers a small area in the prepared susceptibility map (Table 5).

**Referee Comment 38:**
**1) Pg. 15: How are the values for false and true positives calculated? What do those represent in your data?**

2) Author's response:
The description of the tool was extended in the data and methods section.

3) Author's change in manuscript:
The AUC - ROC describes the capability of the model to discriminate between two classes of objects (binary classifiers) i.e. presence or absence of landslides (Chung and Fabbri 2003, Begueria 2006, Frattini et al., 2010). The ROC curve is based on two parameters, namely the true positive rate (TPR) and the false positive rate (FPR). TPR or Sensitivity is calculated using the true positive (TP) and false negative (FN) pixels which are landslides within the classes above and below a threshold value. Conversely, FPR or Specificity is calculated using true negative (TN) and false positive (FP) pixels which are stable pixels within the classes below and above the value threshold, respectively. Thresholds are defined so that the value of the first threshold is lower than the minimum susceptibility observed in the least susceptible class, and the value of the last threshold is higher than the maximum susceptibility in the most susceptible class (Vakhshoori, Zare, 2018). The Area Under the Curve (AUC) measures the performance of the model across all classification thresholds and varies between 0 and 1. The closer to unity AUC is the better the performance of the model, while a value of 0.5 indicates a model that makes random guesses and has no capability to discriminate between the presence or

absence of landslides. Since landslide inventories are rarely complete, the tool was tested in an area where most debris flows had been mapped. The methodological workflow is illustrated in Fig. 3 and explained in the following sections.

**Referee Comment 39:**
**1) Pg. 15, Table 5: Give (Frequency Ratio) equation.**

2) Author's response:
Frequency Ratio equation was included in the text.

3) Author's change in manuscript:
The frequency of landslides in a desired class ($FR_i$) is computed as the ratio between the frequency of landslides in the $F_i$ area ($PL_i$) and the frequency of the $F_i$ area ($PF_i$).

$$FR_i = PL_i/PF_i \qquad \textbf{(Eq. 2)}$$

where $PL_i$ is the ratio between the area of landslides in the $F_i$ area and the area of landslides in the study area; and $PF_i$ is the ratio between the area of the $F_i$ area to the entire study area (Li et al., 2017). A frequency ratio $FR_i$ larger than 1 indicates that the $i$th class of factor F ($F_i$) favours the occurrence of landslides while the opposite is indicated for $FR_i$ smaller than 1.

**Referee Comment 40:**
**1) Pg. 15: (AUC value of the ROC curve) needs to be compared to other published values.**

2) Author's response:
No other AUC-ROC analysis results are reported in the literature for this particular study area and process. The values cannot readily be compared with those derived with other models and in other regions, since the employed datasets might also be different. A comparison of the results against the worst (0.5) and best (1) value outcomes was deemed sufficient.

3) Author's change in manuscript:
None.

**Referee Comment 41:**
**1) Pg. 16: "[…] the spatial resolution of the DTM was resampled to a coarser resolution (from 5 m to 50 m) to ensure consistency between spatial datasets". This is important to mention already in the method section. Since the DEM is very important for the detection of source areas, I think that rather the other data should be adapted to the DEM.**

2) Author's response:
The text was amended as suggested. See also response to Comment 3.

3) Author's change in manuscript:
See comment 3.

CONCLUSIONS

**Referee Comment 42:**
**1) Pg. 16: Influence of stream channels and drainage – I cannot see that this parameter is used in the current susceptibility model.**

2) Author's response:
Please see response to Comments 27 - 29.

3) Author's change in manuscript:
Please see response to Comments 27 - 29.

**Referee Comment 43:**
**1) Pg. 16: Usability of the susceptibility map - How are they supposed to use it. The resolution of 50 m gives quite some restrictions.**

2) Author's response:
Given its scale, the debris flow susceptibility map may be used in the prioiritsation of sites for further, more detailed investigations. We advocate its use at national (not site-specific) scale to highlight hot-spots of potential debris flow activity. We have previously worked with companies (owner and infrastructure manager) who currently have no knowledge of where debris flows could occur in the proximity of their assets. Industry representatives have demonstrated interest in the use of such high level prioritisation tools because it allows them to compare where resources and further work may be required over extended areas.

3) Author's change in manuscript:

   Inserted 'overview' in sentence and added final sentence: Although not without some limitations, the debris flow susceptibility model for Great Britain has built on knowledge by Harrison et al. (2006), refining the model and extending its coverage. As such it represents a useful overview tool for policy-makers, developers and engineers, and can support regional or national scale development action plans and disaster risk reduction strategies at the national scale. Infrastructure owners have expressed interest to BGS in the use of such high level prioritisation tools because it allows them to compare where resources and further work may be required over extended areas.